# Identification of two-dimensional layered dielectrics from first principles

Mehrdad Rostami Osanloo[1], Maarten L. Van de Put[2], Ali Saadat[2] & William G. Vandenberghe [2✉]

To realize effective van der Waals (vdW) transistors, vdW dielectrics are needed in addition to vdW channel materials. We study the dielectric properties of 32 exfoliable vdW materials using first principles methods. We calculate the static and optical dielectric constants and discover a large out-of-plane permittivity in GeClF, PbClF, LaOBr, and LaOCl, while the in-plane permittivity is high in BiOCl, PbClF, and TlF. To assess their potential as gate dielectrics, we calculate the band gap and electron affinity, and estimate the leakage current through the candidate dielectrics. We discover six monolayer dielectrics that promise to outperform bulk $HfO_2$: HoOI, LaOBr, LaOCl, LaOI, $SrI_2$, and YOBr with low leakage current and low equivalent oxide thickness. Of these, LaOBr and LaOCl are the most promising and our findings motivate the growth and exfoliation of rare-earth oxyhalides for their use as vdW dielectrics.

[1] Department of Physics, The University of Texas at Dallas, Richardson, TX, USA. [2] Department of Materials Science and Engineering, The University of Texas at Dallas, Richardson, TX, USA. ✉email: william.vandenberghe@utdallas.edu

Van der Waals (vdW) layered materials such as Transitional Metal Dichalcogenides (TMDs) have been the subject of an enormous amount of research in the last decade[1–3]. The appeal of vdW materials is the natural termination of each layer. Under ideal conditions, a monolayer of a vdW material realizes a "2D material", i.e., a material with thickness less than a nanometer but with a width and length extending over several microns. Having perfect uniformity of the thickness eliminates the severely detrimental effects of surface roughness seen in non-vdW materials when scaled down to sub-nanometer thickness[4]. The natural termination also ensures the absence of surface states traversing the electronic band gap.

The field of nanoelectronics naturally invites the use of 2D materials since a reduction of the channel length and, more recently, channel thickness has been a driver for dramatic technological progress[5,6]. Moreover, surface states have limited the performance and reliability in many nano-electronic applications[7–9] whereas the naturally passivated surfaces of vdW materials alleviate the concern of surface states. As a result, TMDs are now actively being considered as channel materials by the semiconductor industry[10]. High mobilities are reported[3], doping techniques[11] are under development, metal-oxide-semiconductor field-effect transistors (MOSFETs) are being fabricated[12,13], and contact technology is under investigation[11]. TMDs are thus well on the way to commercial application in transistors, being investigated as a replacement of silicon in the active switching devices ("front-end") of semiconductor technology but also as an augmenting technology in the metallization layers that interconnect the devices in the "back-end"[14].

Suitable gate dielectric materials are critical components that allow the "gate" to exercise electrostatic control of the "channel" where electrons flow. The dielectric blocks current flow between the gate and channel (gate leakage) and enhances the electrostatic displacement field (electrostatic control). However, the selection of gate dielectrics to use for vdW materials has not received as much attention. Most TMD-based MOSFETs investigated to date use atomic-layer deposited (ALD) oxides like $HfO_2$ and $Al_2O_3$[15–17]. Unfortunately, when an ALD oxide is deposited on a 2D material, the naturally terminated surfaces now become a large drawback because covalent bonds between the oxide and the 2D material are hard to make[18]. Non-uniform nucleation will give rise to a non-uniform thickness and, in the absence of a uniform thin dielectric, MOSFET performance will become unacceptably poor. One proposed solution addressing the non-uniformity is the deposition a perylene-tetracarboxylic dianhydride molecular crystal layer[19]. Moreover, where covalent bonds are formed, the natural 2D material termination is broken and the surface states we wanted to avoid are reintroduced. It is thus hard to foresee how ALD oxides can ever be a component of a 2D material MOSFET technology.

h-BN is a vdW material that has successfully been used as a dielectric in transistors, in combination with vdW channel materials[5]. However, h-BN also has significant drawbacks such as a low dielectric constant, which is undesired, and the requirement to either transfer h-BN or to grow at high temperatures that are not compatible with semiconductor technology[20,21]. Recently, the crystalline dielectric $CaF_2$ has also been investigated to avoid the drawbacks of amorphous oxides[22,23]. Nevertheless, unlike $Bi_2SeO_5$, which is a layered material, $CaF_2$ is not a layered compound with unterminated bonds at the surface, raising questions of passivation to eliminate interface states and ensure reliability. $Bi_2SeO_5$ does present an interesting native layered oxide although the thickness of a single-layer $Bi_2SeO_5$ (~ 11.47 Å) is significantly thicker than most 2D materials.

In this paper, we use first principles calculations to identify novel vdW dielectric materials. We determine three critical properties for transistor dielectrics: the dielectric constant, the band gap, and the electron affinity. Specifically, we calculate the macroscopic in-plane and out-of-plane dielectric constants of the bulk and monolayer of 32 novel vdW materials using density-functional theory (DFT). We calculate the electron affinity and band gap of the monolayers using hybrid functionals. We model the performance of each vdW material as a gate dielectric, considering its equivalent oxide thickness (EOT) as well as leakage current. To ensure that the materials under consideration are exfoliable[24] or can be grown in monolayer form, we compute the exfoliation energies. We find promising bulk and monolayer materials with high in-plane and out-of-plane dielectric constants for application in n-MOS and p-MOS technologies.

## Results

For a good gate dielectric, a high barrier for electrons or holes, measured by the valence and conduction band offset, is required. The high barrier is required so a dielectric can serve its main purpose, i.e., stopping current flow from the gate to the channel. In addition, a competitive gate dielectric also needs a high dielectric constant, i.e. a "high-k" dielectric, to realize maximal capacitive coupling[25,26]. To first order, the conduction band offset can be approximated as the difference between the affinity of the channel and the affinity of the dielectric while the valence band offset is the conduction band offset augmented by the difference in band gap. So, to identify promising novel dielectrics, reliable estimates of the electron affinity, the band gap and the dielectric constant are required.

A natural ground for exploration of novel materials is presented by the recently developed materials databases Materials Cloud[27], Materials Project[28], and AFLOW[29]. These materials databases contain information on about three and half million inorganic and organic materials whose properties are calculated from first principles using DFT calculations. For our purposes, Materials Cloud is of particular interest as it identifies 457 layered, i.e., "two-dimensional" materials, from bulk materials.

The databases contain a DFT estimate of the band gap and the dielectric constant but unfortunately, the reliability of the DFT estimate used in high-throughput calculations is limited. On the one hand, band gaps are severely underestimated when using the local-density approximation (LDA) or the generalized gradient approximation (GGA). On the other hand, the determination of the dielectric constant using DFT requires a much more stringent convergence criterion compared to calculations of the lattice constants or band gaps, which are usually of interest. So, while indexing a wide variety of materials, the databases use a GGA functional for the band gap calculation, which will significantly underestimate experimental values, electron affinities are not computed, and insufficient precision is used in the calculations to accurately determine the dielectric constant.

To gain a reliable estimate, we calculate the band gaps using hybrid functionals, we compute the electron affinity of a monolayer, and we calculate the dielectric constant with high precision for both bulk and monolayers (see "Methods" section). Of course, using these more advanced calculations, the computational burden increases. So, instead of going through the entire database of 2D materials, we narrow down our search to materials that show promise.

Our selection procedure starts from the 457 materials in Materials Cloud. We consider binary and ternary 2D compounds and select only materials with a band gap exceeding 2.5 eV as calculated using Perdew–Burke–Ernzerof (PBE) GGA functionals in Materials Cloud. The goal of the 2.5 eV criterion is to identify materials that have a band gap 4–6 eV after correcting for the PBE underestimation. We then identify the 3D parent materials

of the selected materials from Materials Cloud, selecting some additional bulk compounds with the same parent material from Materials Project.

Before proceeding with calculations, we prune the dataset through manual inspection. We exclude LiBH$_4$ since it is a deliquescent solid-state material (melting point of 275 °C) at ambient conditions and is highly sensitive to water and oxygen[30]. We also remove NaCN because of its toxic and corrosive properties and its danger to the environment. RbCl is another material we remove from our dataset since, while Materials Cloud identifies it as a potential 2D material, the bulk does not present a layered structure. LiOH, NaOH, Mg(OH)$_2$, and Ca(OH)$_2$ are also eliminated since they are elemental bases, which are very soluble in water and invariably appear as water complexes. We also did not recalculate the values for h-BN as the dielectric properties have been reported accurately and in detail previously[20]. After imposing the >2.5 eV PBE band gap criterion, pruning the aforementioned materials and adding some similar compounds (i.e. space group, crystal system) discovered on AFLOW and Materials Project, we obtain a list containing 32 vdW materials, all of which are halides.

In Fig. 1, we illustrate the 32 vdW materials under consideration and divide the materials into four main categories based on their chemical compositions, space groups, and lattice structures (Supplementary Table 1). Overall, 23 different elements of the periodic table appear: Halogens (F, Cl, Br, I), transition metals (Sc, Ti, Zn, Y, Zr, and Cd), post-transition metals (Al, Bi, In, Sn, Pb), semiconductors (Ge), and Lanthanides (La, Nd, Ho, Lu). Category 1a has a tetragonal lattice with a central layer of halogen/oxygen flanked by metal[31]. Category 1b has an outer halogen layer added to the metal[32–37] and category 1c is category 1b but stretched into an orthorhombic structure. Categories 2, 3, and 4 have a central metal and respectively have a trigonal, tetragonal, and orthorhombic unit cell[38–44].

To ensure all materials are in fact layered, we first calculate the exfoliation energies, $E_{ex}$. The exfoliation energy ranges from 3.12 meV/A$^2$ PbI$_2$ (Cat. 2) to 40.22 meV/A$^2$ LaOCl (Cat. 1b) and the values for each material are listed in the Supplementary Information (Supplementary Fig. 1). As a rule-of-thumb, materials with $E_{ex} < 100$ meV/$^2$ are considered easily exfoliable compounds[24]. Using this criterion, all materials under consideration are layered and exfoliable. Moreover, we calculate the phonon energies of monolayer and bulk for all materials to

investigate the stability of our monolayers. We list the value of monolayer and bulk form phonon energy in Supplementary Tables 9 and 10, respectively. The phonon energy calculation shows that all monolayer materials, except TlF and GeClF, are predicted to be stable.

Figure 2 shows the macroscopic in-plane and out-of-plane dielectric constants for the monolayer of 32 different layered materials. The corresponding values are listed in Table 1. For category 1c, the dielectric constants in the plane are anisotropic and values are provided in Supplementary Table 2. The static dielectric constant ($\varepsilon_0$) includes both the electronic and the ionic contributions to the dielectric response, whereas the optical dielectric constant ($\varepsilon_\infty$) only contains the electronic response. We calculate both the in-plane (||) and out-of-plane (⊥) values of the dielectric constants. To calculate the monolayer dielectric constant, we isolate monolayers in a computational supercell including sufficient vacuum and then rescale the calculated dielectric constants of the supercell to those of the monolayer as done in ref. [34] and discussed in the "Methods" section. Our calculations show that, for monolayers, the highest and the lowest static in-plane dielectric constants belong to TlF (98.4) and GeClF (5.9), while LaOCl (55.8) and MgCl$_2$ (3.56) exhibit the highest and the lowest out-of-plane static dielectric constants, respectively.

Table 1 reveals that, in general, the optical dielectric constant is significantly lower than the corresponding static dielectric constant, indicating a large ionic contribution to the dielectric response for all materials under consideration. Considering the out-of-plane direction, the optical dielectric constant ($\varepsilon_{\infty,\perp}$) ranges from 2.4 (SnF$_4$) to 4.9 (BiOCl) for bulk, and from 2.8 (MgCl$_2$) to 5.8 (CaHI) for monolayers. In contrast, the static dielectric constant ($\varepsilon_{0,\perp}$) is as high as 28.2 for bulk PbClF and 55.8 for monolayer LaOCl. In Supplementary Table 3, we list the experimentally determined values of the dielectric constant for CdBr$_2$, CdCl$_2$, and PbI$_2$, and find agreement within 20% between the theoretical and experimental values[43–45].

Compared to the dielectric constants of 3D "high-k" oxides Al$_2$O$_3$ (9), Y$_2$O$_3$ (15), ZrO$_2$ (25), and HfO$_2$ (25);[46,47] bulk BiOCl, GeClF, PbClF, LaOBr, LaOCl, SrHBr, and TlF offer high out-of-plane static dielectric constants, ranging from 11.1 (SrHBr) to 28.2 (PbClF). The large difference between the in-plane and out-of-plane static dielectric constants comes from the ionic contribution, as it is not observed at optical frequencies. Indeed, the

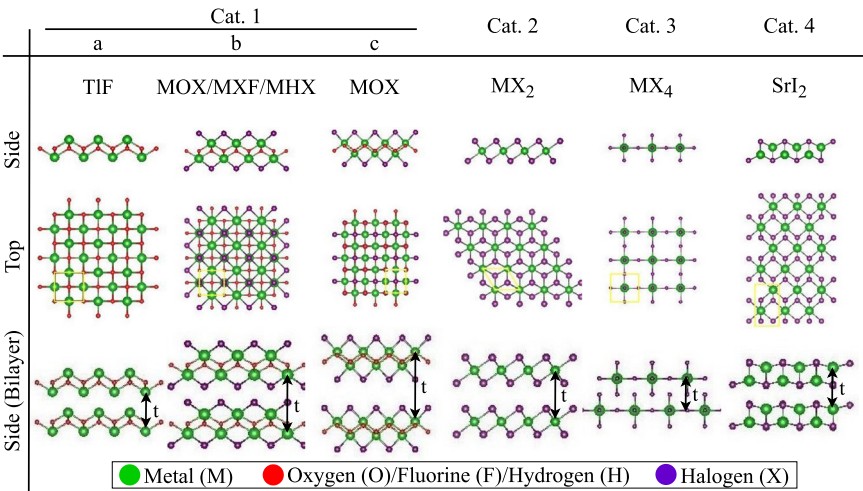

**Fig. 1 The structure of 32 vdW materials.** Side and top views of the monolayer structures are shown, where the yellow squares represent the computational unit cells. Side views of the bilayer structures are also presented, showing the A–A and A–B stacking configurations. The measurement of the thickness of the monolayers ($t$) is indicated. Category 1 contains three similar structures, further divided in subcategories a, b and c.

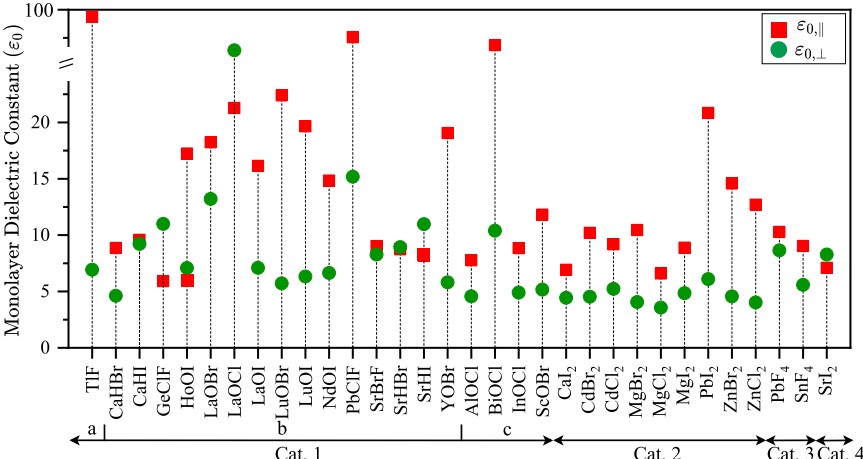

**Fig. 2 The monolayer static in-plane and out-of-plane dielectric constants of 32 layered vdW materials.** Red squares and green circles respectively denote the in-plane and the out-of-plane dielectric constants. The monolayer in-plane dielectric values range from 5.9 (GeClF) to 98.4 (TlF), and the out-of-plane dielectric values range from 3.6 (MgCl$_2$) to 55.8 (LaOCl).

**Table 1 In-plane and out-of-plane macroscopic dielectric constant of monolayer (1 L) and bulk of 2D layered materials.**

|  | Bulk $\varepsilon_{\infty,\perp}$ | Bulk $\varepsilon_{\infty,\parallel}$ | Bulk $\varepsilon_{0,\perp}$ | Bulk $\varepsilon_{0,\parallel}$ | 1 L $\varepsilon_{\infty,\perp}$ | 1 L $\varepsilon_{\infty,\parallel}$ | 1 L $\varepsilon_{0,\perp}$ | 1 L $\varepsilon_{0,\parallel}$ |
|---|---|---|---|---|---|---|---|---|
| TlF (1a) | 4.1 | 4.2 | 8.8 | 63.9 | 3.6 | 3.6 | 6.9 | 98.4 |
| CaHBr (1b) | 3.8 | 4.0 | 7.0 | 8.8 | 4.1 | 3.9 | 4.6 | 8.9 |
| CaHI (1b) | 4.4 | 4.6 | 6.7 | 9.1 | 5.8 | 4.7 | 9.2 | 9.6 |
| GeClF (1b) | 4.3 | 6.0 | 18.4 | 6.0 | 4.3 | 5.9 | 11.0 | 5.9 |
| HoOI (1b) | 4.3 | 5.1 | 6.1 | 16.7 | 4.6 | 5.0 | 7.1 | 17.2 |
| LaOBr (1b) | 4.6 | 4.9 | 12.5 | 18.4 | 5.3 | 4.7 | 13.2 | 18.2 |
| LaOCl (1b) | 4.2 | 4.4 | 11.8 | 18.4 | 5.5 | 4.5 | 55.8 | 21.3 |
| LaOI (1b) | 4.4 | 5.1 | 7.3 | 16.7 | 4.9 | 4.9 | 7.1 | 16.1 |
| LuOBr (1b) | 3.7 | 4.4 | 5.8 | 20.7 | 4.0 | 4.5 | 5.7 | 22.4 |
| LuOI (1b) | 4.2 | 5.2 | 6.1 | 19.0 | 4.5 | 5.1 | 6.3 | 19.7 |
| NdOI (1b) | 4.3 | 5.1 | 6.5 | 14.6 | 4.8 | 5.0 | 6.6 | 14.8 |
| PbClF (1b) | 4.2 | 4.8 | 28.2 | 56.2 | 4.6 | 5.0 | 15.2 | 74.6 |
| SrBrF (1b) | 3.3 | 3.5 | 7.9 | 6.8 | 3.8 | 3.3 | 8.3 | 9.0 |
| SrHBr (1b) | 4.0 | 3.9 | 11.1 | 8.8 | 5.2 | 3.9 | 8.9 | 8.7 |
| SrHI (1b) | 4.2 | 4.3 | 7.9 | 8.3 | 4.8 | 4.2 | 11.0 | 8.2 |
| YOBr (1b) | 3.8 | 4.4 | 6.3 | 18.1 | 4.1 | 4.4 | 5.8 | 19.1 |
| AlOCl (1c) | 3.0 | 3.1 | 4.4 | 7.8 | 3.1 | 3.1 | 4.6 | 7.8 |
| BiOCl (1c) | 4.9 | 6.5 | 11.7 | 56.0 | 5.3 | 6.4 | 10.4 | 64.9 |
| InOCl (1c) | 3.4 | 3.8 | 4.9 | 8.9 | 3.6 | 3.7 | 4.9 | 8.9 |
| ScOBr (1c) | 3.8 | 4.2 | 5.0 | 11.9 | 3.9 | 4.1 | 5.2 | 11.8 |
| CaI$_2$ (2) | 3.5 | 3.9 | 4.2 | 6.8 | 3.7 | 3.8 | 4.4 | 6.9 |
| CdBr$_2$ (2) | 3.6 | 4.3 | 4.5 | 10.0 | 3.7 | 4.3 | 4.5 | 10.2 |
| CdCl$_2$ (2) | 3.7 | 3.3 | 4.0 | 8.5 | 3.7 | 3.8 | 5.2 | 9.2 |
| MgBr$_2$ (2) | 3.2 | 3.5 | 3.9 | 7.6 | 3.2 | 3.5 | 4.1 | 10.5 |
| MgCl$_2$ (2) | 2.7 | 3.0 | 3.5 | 6.6 | 2.8 | 2.9 | 3.6 | 6.6 |
| MgI$_2$ (2) | 3.9 | 4.4 | 4.6 | 8.8 | 4.0 | 4.4 | 4.9 | 8.9 |
| PbI$_2$ (2) | 4.9 | 6.8 | 5.7 | 25.7 | 5.0 | 6.3 | 6.1 | 20.8 |
| ZnBr$_2$ (2) | 3.7 | 4.6 | 4.5 | 15.3 | 3.7 | 4.6 | 4.6 | 14.6 |
| ZnCl$_2$ (2) | 3.1 | 3.8 | 3.9 | 12.3 | 3.1 | 3.8 | 4.0 | 12.7 |
| PbF$_4$ (3) | 2.9 | 6.6 | 4.8 | 12.8 | 4.0 | 2.9 | 8.6 | 10.3 |
| SnF$_4$ (3) | 2.4 | 5.5 | 3.9 | 10.9 | 3.0 | 2.4 | 5.6 | 9.0 |
| SrI$_2$ (4) | 3.8 | 3.8 | 7.3 | 6.9 | 5.1 | 3.9 | 8.3 | 7.1 |

$\varepsilon_\infty$ and $\varepsilon_0$ are the electronic and the static ("electronic" + "ionic") components of the dielectric constants. Both out-of-plane (⊥) and in-plane values (∥) are reported. Materials category 1a–4, illustrated in Fig. 1 and space groups provided in Supplementary Table 1, are indicated in parenthesis in the first column for each material.

ionic response is governed by the strength of the covalent bonds in the in-plane versus the weak vdW bonding in the out-of-plane direction information (Supplementary Fig. 2).

Going from bulk to monolayer, we first compare the dielectric response at optical frequencies. We observe that the in-plane response ($\varepsilon_{\infty,\parallel}$) changes by less than 15%, except for the Cat. 3 materials: PbF$_4$ (−56%) and SnF$_4$ (−57%). Turning out-of-plane, we find a range of materials for which the optical dielectric response ($\varepsilon_{\infty,\perp}$) increases significantly: CaHI, LaOBr, LaOCl, SrHBr, PbF$_4$, SnF$_4$ and SrI$_2$, by up to 39% for PbF$_4$. Next, we include the ionic response at low frequencies and look closer at the change in static dielectric response when going from bulk to monolayer. While the in-plane response ($\varepsilon_{0,\parallel}$) for most materials does not change significantly; TlF, NdOI, PbClF, and MgBr$_2$ show an increase between 33% and 54% which is caused by an increased ionic response in their monolayer form. In the out-of-plane direction ($\varepsilon_{0,\perp}$), the ionic contribution further increases the dielectric response for monolayer PbF$_4$ and SnF$_4$, compared to their bulk form. In contrast, the out-of-plane ionic response in CaHBr, GeClF, PbClF, and SrHBr is suppressed by up to 50% compared to their bulk forms.

Finally, note that compared to their bulk forms, monolayer LaOBr, LaOCl, SrBrF, and SrI$_2$ are unique in showing a significantly improved out-of-plane electronic dielectric response, while they do not see a significant change in their ionic response. Of these materials, LaOBr has the third-highest static out-of-plane dielectric constant among all monolayers (13.2). Only LaOCl (55.8) and PbClF (15.2) have higher out-of-plane dielectric constants, although the ionic dielectric response in monolayer PbClF is significantly reduced compared to bulk, while it is strongly enhanced in monolayer LaOCl. Based only on their out-of-plane dielectric constants, LaOBr, LaOCl, and PbClF could be good candidates for a gate dielectric, if they turn out to be good insulators as well.

To investigate the insulating properties of the candidate dielectrics, Fig. 3 shows the electron affinity and the band gap, or, equivalently, the conduction and valence band edges, where the vacuum level is set to zero. The red square points on each line indicate the conduction band edge, i.e., minus the electron affinity, while the green circle points indicate the valence band edge. The band gap and its value (in eV) are indicated. We calculate the band gaps and the electron affinity of all materials using the

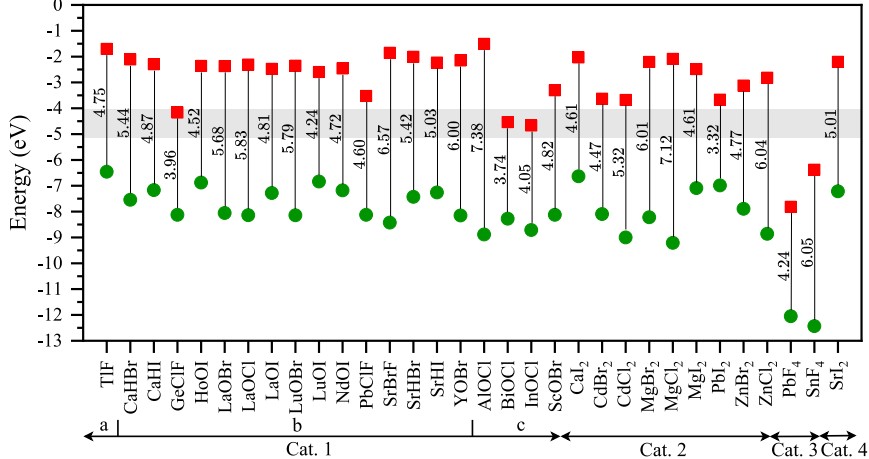

**Fig. 3 Conduction (red square) and valence (green circle) band edge of 32-monolayer material with respect to the vacuum level (0 eV).** The lines indicate the obtained band gap with corresponding value in eV. Values are calculated using a hybrid HSE06 functional. A channel material with a 4 eV affinity and a 1 eV band gap is added as a gray band for reference.

Heyd–Scuseria–Ernzerhof (HSE06) hybrid functional[48,49]. Compared to non-hybrid functional PBE calculations, HSE06 band gap predictions are much more reliable, producing considerably larger band gaps than those obtained from PBE calculations. We report the HSE and PBE band gap values along with the electron affinity values in Supplementary Table 4. Even larger predicted $G_0W_0$ band gaps obtained from other theoretical studies[50] along with experimental band gaps for some 2D monolayers[51,52] are included in Table S5. A good dielectric candidate material must have a band offset exceeding 1 eV with the channel material to minimize leakage current caused by Schottky emission of carriers into the dielectric[53]. Materials such as CaHBr, LaOBr, LaOCl, and SrBrF are examples of wide-gap materials with conduction band offset greater than 1 eV, with respect to a channel with a 4 eV affinity and are promising as dielectrics.

Although there are superior methods available (e.g., GW) compared to hybrid functionals, they come at a significant computational cost. HSE balances computational expense while avoiding the underestimation of the band gap resulting from LDA or GGA. In this work, we only report on the long-wavelength dielectric constant[54]. This strongly reduces the severity of the long-range interactions that may adversely affect supercell methods. Since the Coulomb kernel of a 1D dipole is a step function, it does not have a long-range interaction unlike a 2D or 3D dipole. To show that the long-range interactions are not an issue, we performed the monolayer calculations for LaOCl and LaOBr with three additional different vacuum sizes (see Supplementary Table 6). The sensitivity of the extracted dielectric constant to the one calculated from DFT can be quantified by calculating $\frac{d\varepsilon_{2D}}{d\varepsilon_{sc}}$. In the out-of-plane direction $\frac{d\varepsilon_{2D}}{d\varepsilon_{sc}} = \frac{c}{t}\frac{\varepsilon_{2D}^2}{\varepsilon_{sc}^2}$ while in the in-plane direction $\frac{d\varepsilon_{2D}}{d\varepsilon_{sc}} = \frac{c}{t}$. For LaOBr and LaOCl, we reported the $\frac{d\varepsilon_{2D}}{d\varepsilon_{sc}}$ in the out-of-plane and in-plane directions in Supplementary Table 7. Since $\frac{d\varepsilon_{2D}}{d\varepsilon_{sc}}$ scales with $\varepsilon_{2D}^2$, LaOCl is orders of magnitude more sensitive to error propagation. The extracted dielectric constants do no change significantly except for the ionic response in LaOCl where the error propagation in the DFPT calculation affects the results. Nevertheless, these results indicate that environmental screening by periodic images does not affect the obtained dielectric constants.

The ideal dielectric is one with a small thickness, high dielectric constant, and small leakage current[47,55]. To quantitatively measure the promise of a gate dielectric material for *n*-MOS and *p*-

MOS applications, we compute the leakage current due to Fowler–Nordheim tunneling and thermionic emission[56,57] using "standard" equations as detailed in the "Methods" section and Supplementary information (Supplementary Table 8). To identify the most promising materials, we compare EOT versus leakage current, where we assumed an *n*-MOS with a 4 eV electron affinity for the channel material. We make a similar analysis for *p*-MOS (Supplementary Fig. 3 and Supplementary Table 9) but this reveals that *p*-MOS leakage current is unlikely to be an issue for any of the materials under consideration, except for $PbF_4$ and $SnF_4$. As a reference, we also compute the leakage current and EOT of single-layer h-BN. Inspecting the International Roadmap for Devices and Systems (IRDS)[58], we identify a leakage current less of 100 pA/μm as an absolute maximum for any viable gate dielectric.

In Fig. 4, we show the performance of monolayer (1 L), and bilayer materials, compared to $HfO_2$. Since the calculation of multilayer (bilayer, trilayer, etc.) is computationally expensive, we estimate the bilayer performance based on the dielectric constant of the monolayer (filled squares) and bulk (hollow squares) material, connected by a line. This representation shows the uncertainty in the dielectric response of the bilayer in our estimate. We show the calculated EOT with respect to the calculated leakage current for an *n*-MOS. Materials closer to the lower left are better gate dielectrics, featuring low EOTs with low leakage currents. We identify several monolayer and bilayer materials, which outperform $HfO_2$ with a 0.4 nm interfacial layer of $SiO_2$[59]. Moreover, we identify 8 monolayer dielectrics that outperform pure bulk $HfO_2$: HoOI, LaOBr, LaOCl, LaOI, SrHBr, $SrI_2$, TlF, and YOBr. These materials feature leakage current densities ranging from $10^{-52}$ A/cm$^2$ to $10^{-19}$ A/cm$^2$ and an EOT ranging from 0.05 to 0.5 nm.

## Discussion

The most promising is monolayer LaOCl, having the lowest EOT (~ 0.05 nm), by a fair margin, among all materials, and with a leakage current less than $10^{-7}$ A/cm$^2$. Furthermore, even bilayer LaOBr outperforms bulk $HfO_2$ with an EOT < 0.5 nm and leakage currents $<10^{-18}$ A/cm$^2$ making these rare-earth oxyhalides the most promising in our list of 32 materials. Note that while LaOBr is in the Materials Cloud "layered materials" database, LaOCl is not and it is a material we added from the Materials Project database. We limited our search to 32 materials so other

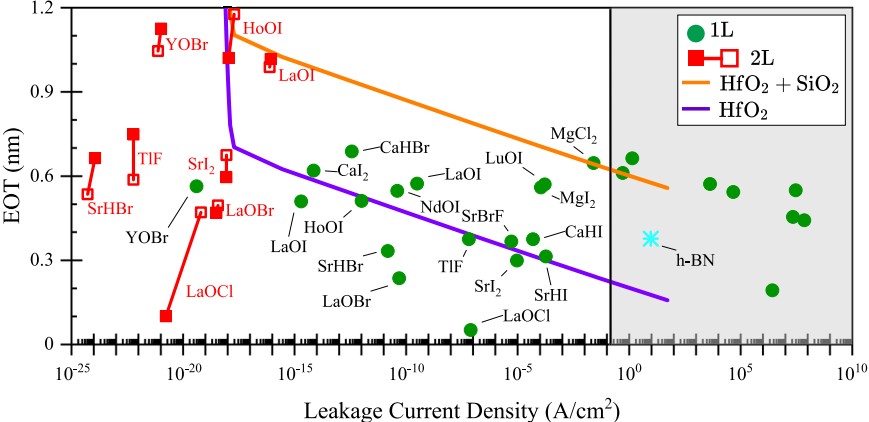

**Fig. 4 Leakage current and EOT for monolayers (green circles) and bilayers (red squares) in *n*-MOS applications.** Reference dielectrics are also shown: The blue star shows monolayer h-BN. The dark purple line shows the EOT corresponding to the different thicknesses of $HfO_2$ (between 1 and 10 nm), while the orange line represents 0.4 nm of interfacial $SiO_2$ in addition to $HfO_2$. The shaded light gray area shows values that fall outside of the IRDS leakage current (per pitch) criterion of 100 pA/μm for a transistor with a 28 nm pitch, an 18 nm long gate, and an effective gate width of 107 nm[58]. The acceptable current density becomes 0.145 $A/cm^2$. For the bilayer, we estimate the performance using the dielectric constant of monolayer (filled squares) and using the bulk dielectric constant (hollow squares).

promising rare-earth oxyhalides not included in our present investigation, like GdOCl or YOCl, may also be promising gate dielectrics to be identified in future investigations.

Recently, two new dielectrics have been proposed for vdW materials, $CaF_2$ and $Bi_2SeO_5$. $CaF_2$ has been shown to have a desirable dielectric constant of 8.4, and an enormous band gap of 12.1 eV[22]. The thermally stable $Bi_2SeO_5$ has been demonstrated with a dielectric constant of 21 and a moderate band gap of 3.9 eV[23]. However, while both $CaF_2$ and $Bi_2SeO_5$ outperform other bulk dielectrics such as $HfO_2$, $CaF_2$ is not a vdW material and is prone to the same surface roughness and interface defects of conventional oxides. It thus remarkable that we have identified eight monolayer vdW materials that outperform $HfO_2$, the industry-leading bulk dielectric, without even considering the intrinsic benefits of vdW dielectrics, e.g., perfect interfaces without defects. All eight monolayers exhibit high band gaps (>3 eV), high dielectric constants (>5.8), tiny leakage current (<$10^{-5} A/cm^2$), small EOT (< 0.6 nm), and suitable band offsets.

Our most promising materials LaOBr and LaOCl, which outperform $HfO_2$, $CaF_2$, and $Bi_2SeO_5$, are known stable and readily available materials. LaOBr and LaOCl are water insoluble and have been investigated for applications as scintillators and ion transport channels[35,37]. Previously, LaOBr and LaOCl have been synthesized using a solid-state reaction between Lanthanum Oxide ($La_2O_3$) and ammonium chloride/ammonium bromide ($NH_4Cl/NH_4Br$). We could not find any literature on attempts to exfoliate or characterize monolayers of LaOCl or LaOBr. Our calculations show that they are not just layered but in fact exfoliable and show that LaOBr and LaOCl have the potential to realize highly performant true vdW field-effect transistors. We hope that our result encourages further experimental investigation into the materials we identified (HoOI, LaOBr, LaOCl, LaOI, SrHBr, $SrI_2$, TlF, and YOBr) and specifically into the monolayer and bilayer form of the rare-earth oxyhalides LaOBr and LaOCl.

To conclude, starting from a database of layered materials, we selected 32 viable candidates for suitable vdW dielectric applications (exfoliable, sufficiently large band gap, and stable). For each material, we calculated the in-plane and out-of-plane macroscopic dielectric constants from first principles. Our calculations show a wide range of in-plane and out-of-plane dielectric constants, from 2.5 to 98.4. To gauge the performance of each material as a gate dielectric in *n*-MOS applications, we calculated the leakage current and the EOT for each material. Since

hydrobromides are generally hygroscopic materials and TlF monolayers were found to be unstable, we exclude SrHBr and TlF from the shortlist of candidate materials ending up with six promising vdW dielectrics: HoOI, LaOBr, LaOCl, LaOI, $SrI_2$, and YOBr, all of which promise better performance than $HfO_2$. The best performing material, monolayer LaOCl, shows immense promise as a gate dielectric, with an EOT < 0.1 nm while maintaining leakage currents < $10^{-7} A/cm^2$. Monolayer dielectrics may not be sufficiently robust to defects and in this case, only LaOBr and LaOCl show sub−0.5 nm EOT in their bilayer forms. Furthermore, LaOBr and LaOCl are known and stable materials. We hope that our research leads to the further exploration of rare-earth oxychlorides and oxybromides for applications as layered dielectrics.

## Methods

**Calculation details**. We employ DFT as implemented in the Vienna ab initio simulation package (VASP)[60,61] and adopt the GGA as proposed by PBE[62] for the electron exchange and correlation functional. To ensure high accuracy in our calculation, we increase the plane-wave energy cutoffs by at least 30% compared to their recommended minimum value. For the 2D mono-, bi-, and trilayers, we use supercells with sufficient vacuum in the *z* direction, measuring at least 15 Å (Supplementary Table 10).

To obtain precise and reliable dielectric values, we set the energy convergence criteria to $10^{-8}$ eV and relaxation is performed until the force on each atom is less than $10^{-3}$ eV Å$^{-1}$. To sample the Brillouin Zone (BZ), $12 \times 12 \times 12$ and $12 \times 12 \times 1$ k-point grids are used for the bulk and the few-layered structures, respectively. To account for vdW interactions, we use the DFT-D3 method of Grimme's[63]. Finally, the exfoliation energy is extracted as a difference between the ground state energies of bulk and monolayers[64,65].

**Dielectric tensor of bulk**. We employ density-functional perturbation theory (DFPT), as implemented in VASP, to calculate the permittivity tensor of the bulk unit cell, from which we extract the in-plane ($\varepsilon_\parallel$) and out-of-plane dielectric constants ($\varepsilon_\perp$). To calculate the in-plane dielectric constant ($\varepsilon_\parallel$) of the materials, we average over the *x* and *y* components, so that $\varepsilon_\parallel = (\varepsilon_x + \varepsilon_y)/2$. The macroscopic out-of-plane dielectric constants are the same in both cases ($\varepsilon_\perp = \varepsilon_z$). From VASP, we extract both optical and static dielectric constants. The optical dielectric constant ($\varepsilon_\infty$) represents the high-frequency response where only the electrons can respond to an applied electric field. The static dielectric constant ($\varepsilon_0$), on the other hand, represents the low-frequency response where both the electrons and ions can respond[66]. We also calculate $EOT = \left(\frac{\varepsilon_{SiO_2}}{\varepsilon_{dielectric}}\right) t_{dielectric}$ to easily compare the performance of various dielectric materials.

**Phonon calculation**. We calculate the phonon energies and vibrational modes for both monolayer and bulk of all materials from DFPT. Phonon energy calculations, acoustic phonons in particular, indicate the stability of a system. We report the

monolayer and the bulk Phonon energies in Supplementary Tables 11 and 12, respectively.

**Vacuum elimination from the 2D structures**. Since VASP implements DFT using a plane-wave basis, the unit cells are periodic in the $x$ and $y$ directions (in-plane) but also in the $z$ direction (out-of-plane). For 2D layers, the supercell dielectric values contain a vacuum contribution that must be removed to analyze the dielectric constant of the layers themselves. To extract the dielectric values of the 2D monolayer structures, we rescale the dielectric constants calculated for the supercells using the same procedure as bulk. Following ref. [20], we use the two following equations:

$$\varepsilon_{2D,\perp} = \left[1 + \frac{c}{t}\left(\frac{1}{\varepsilon_{SC,\perp}} - 1\right)\right]^{-1} \quad (1)$$

$$\varepsilon_{2D,\parallel} = 1 + \frac{c}{t}\left(\varepsilon_{SC,\parallel} - 1\right) \quad (2)$$

where $c$ is the supercell height, and $t$ is the thickness of monolayers. The thickness is extracted from the inter-layer distance of the bilayer as indicated in Fig. 1.

We also studied the sensitivity of our method with respect to vacuum size. While our results are converged for most materials, we found that, for materials with extremely large out-of-plane dielectric constants, (e.g. LaOCl) significant errors are observed (see Supplementary Table 6). Unfortunately, the ionic out-of-plane response does show sensitivity to the vacuum used and is not converged even at the strictest energy threshold. We attribute this to the limitations in accuracy of the DFPT in VASP. The latter is not entirely surprising as the dielectric constant rescaling procedure amplifies errors, and phonon-related quantities, especially those involving polarization vectors, are known to have much larger errors compared to the ground state[67].

**Tunneling current and thermionic emission**. To calculate the leakage current through the semiconductor-metal interface we use the Fowler–Nordheim tunneling current and thermionic emission over the barrier:[56,57]

$$J_{tun} = \frac{q^3 \, \varepsilon^2}{8\pi h \varphi} \exp\left(-\frac{4\sqrt{2m^*} \, \varphi^{3/2}}{3qh\varepsilon}\right) \quad (3)$$

$$J_{therm} = A^{**} \, T^2 \exp\left(\frac{-q\left(\varphi - \sqrt{q\varepsilon/(4\pi\varepsilon_i)}\right)}{kT}\right) \quad (4)$$

where $\varepsilon$, $\varphi$, $\varepsilon_i$, $A^{**}$, $T$, $q$, $m^*$, and $k$ are the electric field in the insulator, barrier height, insulator permittivity, effective Richardson constant, temperature, electron charge, electron effective mass, and Boltzmann constant, respectively. The electric field ($\varepsilon$) in the equations above is the ratio of the applied voltage to the dielectric thickness, $t$. The total leakage current is given by the sum of the tunneling and the thermionic current, $J_{tot} = J_{tun} + J_{therm}$.

We use a 0.7 V power supply voltage, $V_{dd}$, and 345 mV saturation voltage, $V_{sat}$, taken from IRDS[58] and yielding an applied voltage of 355 mV for setting an appropriate criterion for the leakage current and the calculation of electric field in the insulator. For the monolayers, we use the electron effective mass as the out-of-plane tunneling mass. For HfO$_2$, we use a tunneling mass of 0.11m$_e$[68], hole effective mass of 0.58m$_e$[69], band gap of 6 eV and electron affinity of 2.5 eV[70].

To estimate leakage current, we calculate the out-of-plane effective masses from the energy dispersion diagram $(E–K)$ across the conduction band minimum (for the electron effective mass) or the valence band maximum (for the hole effective mass). We extract the effective mass by computing a 100 k-point path in the bulk band structure using the PBE functional. The k-point path traverses the BZ in the out-of-plane direction and is chosen to start at the band extremum and to end at the BZ edge. The effective mass is extracted from the band curvature along the out-of-plane direction using:

$$\frac{1}{m^*} = \frac{1}{\hbar^2}\frac{d^2 E(k)}{dk^2} \quad (5)$$

where $E(k)$ is the energy of the carrier at wavevector $k$, varying in the out-of-plane direction, and $\hbar$ is the reduced Plank constant.

## Data availability
The input files used in this study have been deposited in the NOMAD repository database under accession code (https://doi.org/10.17172/NOMAD/2021.07.18-1). The processed dielectric data are available in the main paper. The lattice constant, band gap and calculated leakage current are provided in the Supplementary Information file.

## Code availability
The codes that are necessary to reproduce the findings of this study are available from the corresponding author upon reasonable request. All DFT calculations were performed by using the Vienna ab initio simulation package (VASP).

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

## Acknowledgements

The project or effort depicted was or is sponsored by the Department of Defense, Defense Threat Reduction Agency. The content of the information does not necessarily reflect the position or the policy of the federal government, and no official endorsement should be inferred.

## Author contributions

M.R.O. and W.G.V. conceived the project. M.R.O. developed the code and performed the simulations. M.L.V.de.P., A.S. and W.G.V. analyzed the obtained results. M.R.O. and A.S. wrote the paper with M.L.V.de.P. and W.G.V. contributing to the discussion and preparation of the manuscript.

## Competing interests

The authors declare no competing interests.
