## [Peer Review File · Nature Communications]

Identification of Two-Dimensional Layered Dielectrics from First PrinciplesREVIEWER COMMENTS

Reviewer #1 (Remarks to the Author):

In this work, a variety of large bandgap vdW materials were calculated to show their probable dielectric properties for potential usage as dielectric free of interface scattering in 2D electronics. Their out-of-plane dielectric constant, bandgap, electron affinity and leakage current were calculated, and 7 promising 2D dielectric materials were picked out. This work offered a significant guidance of the development of dielectrics with vdW surface for 2D electronics. Hence, the reviewer would recommend the publication in Nature Communications if the authors can address the following issues appropriately, especially the first one:

Main issue 1:

The benchmarking of these materials with the former reported ones cannot be carried out, as the evaluation criterion of the leakage current of the dielectrics must be unified:

The unit of leakage current density J is A/cm^2 intrinsically (and obtained directly), rather than A/cm or $pA/\mu m$. Please show the leakage current J count by A/cm^2 in the figures for a comparison with previous works. Note that the leakage current count by $pA/\mu m$ is strongly dependent on the device size and structure, but no details of the device model used was shown. Besides, please double check whether the unit is misused.

The leakage current should be calculated at the gate voltage of 1V other than 0.7V in order to make it comparable with previous works.

The effective mass, m^* , has great influence on J_{tunnel} , and differs a lot between materials. Effective mass can be extracted from the band structure of CBM and VBM, obtained alongside with the calculation of bandgap. If possible, make a list of the calculated m^* , and recalculate the leakage current with them.

These two articles for the evaluation criterion of the leakage current of gate dielectrics:

(John Robertson & Robert M. Wallace, Materials Science and Engineering R, 2015, 88, 1–41, especially Fig.2 and Fig.9; Yury Yu. Illarionov et al. Nature Communications, 2020, 11,3385, especially Fig.3). These references repay careful reading.

Other issues:

1. Article Ref. 32 was cited one-sidedly in page 2. The article offered the method to avoid the problem of non-uniformity with a PTCDA buffer layer. The author can find its shortages such as a thicker EOT, but should not miss the information it brought about.
2. Some of the materials (especially in cat. 2) are common and their dielectric constant (experimental data) can be found. Please display them in the same chart to reflect the error range of the calculation.
3. In this work, LaOCl was found having an ultrahigh out-of-plane dielectric constant (55.8), far more higher than its bulk (11.8). To ensure the reliability, the dielectric constant of few-layer LaOCl should be provided in supporting information.
4. In page 2, "Nevertheless, neither CaF_2 nor Bi_2SeO_5 are layered compounds with unterminated bonds at the surface, raising questions of passivation to eliminate interface states and ensure reliability". Actually, Bi_2SeO_5 is a layered material without dangling bonds between the gap. The author needs to correct related statements. Besides, single-crystalline, polycrystalline and amorphous Bi_2SeO_5 should be distinguished.
5. There are some minor errors that may be clerical errors. "either transfer h-BN or to grow at temperatures" should be "either transfer h-BN or to grow at high temperatures". (page2, line18) The unit of exfoliation energies should be " $meV/\text{\AA}^2$ " instead of " $eV/\text{\AA}^2$ ". (page4, line14-17)
6. The chemical instability of H- is well-known. Thus, it's better not to recommend the ones with H- in the conclusion.

Reviewer #2 (Remarks to the Author):

In this manuscript, the authors analyze a database of 2D materials and refine the existing calculations in order to find a subset of promising materials as gate dielectrics. The authors use more accurate calculations to improve the estimate of the electronic properties, and analyze

criteria such as dielectric constants, band gaps and leakage currents. The authors identify some 2D materials which could be useful for 2D devices. The results are interesting and I recommend publication after minor revisions.

* At the end of page 3, the authors state "After imposing the >2.5 eV PBE bandgap criterion, pruning the aforementioned materials and adding some similar compounds discovered on AFLOW and Materials Project, ..."

The selection from the Materials Cloud dataset is described clearly. However, the authors should specify what "similar compounds" means and how they have been found. Especially since one of the proposed materials LaOCl is picked from these two databases.

* Figure S1 (probably, I can't read the caption) is truncated at the bottom.

* The in-plane dielectric constants are found as an average of the two cartesian directions $(\epsilon_{x,x} + \epsilon_{y,y})/2$. However, are the materials isotropic? If not, what is the degree of anisotropy? This consideration is likely important for manufacturing devices.

Reviewer #3 (Remarks to the Author):

In the manuscript "Identification of Two-Dimensional Layered Dielectrics from First Principles" Osanloo et al. present a systematic first principles study of novel layered materials with a focus on their potential as gate dielectrics as needed for field-effect transistor devices. The authors extract 32 promising candidates from material databases and calculate their dielectric properties (in bulk and monolayer), electron affinities, and band gaps. From this they identify seven candidates which outperform HfO₂.

The presented study is interesting and the manuscript is well written and certainly merits publication, but I'm afraid that it is in its present form not suitable for the broad readership of Nature Communications due to two main reasons:

1.) The manuscript is not very accessible for non-expert readers who are not familiar with the functional basics of field-effect transistor devices and their realizations using layered materials. As an example the abstract and the introduction (more or less) directly start with the discussion of "channel" and "gate" materials without explaining their individual roles, while the title is rather general. Also, the current role of HfO₂ is not entirely clarified so that the reoccurring comparison to it lacks context.

2.) There are a variety of methodological questions open (see below). Without the corresponding clarifications it is not clear how reliable the ab initio results are.

The following first lists the open methodological questions followed by a list of minor remarks.

A) It is well known that due to reduced "environmental screening" in layered materials (in the bulk and especially in the monolayer) these materials hosts enhanced long-range Coulomb interactions, which decisively affect (enhance) the band gaps of layered semi-conductors. These effects are not captured in conventional LDA or GGA based DFT calculations and can only be properly taken into account within many-body perturbation theory in form of GW calculations (see e.g. Phys. Rev. Lett. 111, 216805, 2013 or Phys. Rev. B 94, 155406, 2016). As an approximation HSE functionals might be used to enhance the LDA/GGA band gaps, but it is a priori not clear how well HSE performs for specific layered materials and whether the same amount of exact exchange inter-mixing is needed for the monolayer and bulk systems. Since the authors used here the HSE functional for the monolayer band-gap calculations it is not clear how reliable these values are. I would suggest to perform at least for one candidate from each category a G₀W₀ calculation to check and define the HSE accuracy for the specific monolayers.

B) The authors study the ability to exfoliate monolayers for each material by calculating the exfoliation energies from bulk and monolayer calculations. I see two problems with this:

B-1) Again it is not clear to me how reliable these exfoliation energies as gained from GGA+Grimme correction are in comparison to full RPA calculations.

B-2) The ability to exfoliate monolayers from bulk vdW materials should to my mind not just be defined by the exfoliation energy but also by the monolayer stability. The latter can, for example, be studied by calculating the phonon dispersions from DFPT. This should always be checked when novel monolayer materials are predicted and investigated.

C) Based on the comment from B-2) I also wonder how strong the in-plane ϵ_0 of TIF, PbClF, and BiOCl in bulk and monolayer might be influenced by (very) soft phonons?

D) It looks like that for the calculation of the dielectric properties the PBE+Grimme exchange functional has been used instead of the HSE one as used for the bandgap calculations in the monolayer limits. Since the value of the band-gap affects also the dielectric properties of a semiconductor the authors should either compare the results between PBE and HSE or should at least comment on this point.

C) The accuracy of the "Vacuum elimination [in the dielectric properties] from the 2D structures" is not entirely clear to me. From a conventional RPA point of view, in which the dielectric function is defined by $\epsilon = 1 - V \cdot \chi$ with V being the bare Coulomb interaction and χ the bare susceptibility, it is clear that the long-range V is responsible for spurious interlayer screening even for large supercells (χ is rather localized), which needs to be corrected for. For GW-like calculations this has been discussed for example in Phys. Rev. B 94, 155406, 2016 or Phys. Rev. B 92, 2015. Here and in Ref. 34, the authors use DFPT based on PBE to calculate ϵ . Thus, I wonder if the applied "vacuum elimination" solves the same problems here as indicated from the RPA example above.

Minor points:

I) What is meant with "back-end" and "front-end" on page 2?

II) It is not clear on which ab initio level the mono-, bi-, and tri-layer data shown in Fig. 4 has been calculated.

III) With respect to the strongly (in plane) anisotropic screening properties of black phosphorus, it might be interesting to also give ϵ_x and ϵ_y separately and not just their average.

IV) For the "vacuum elimination" the layer-layer distance is needed. Again, it is not clear how this has been evaluated, i.e. which exchange-correlation functional has been used and how reliable this is.

We thank the reviewers for their time and consideration. Below is a point-by-point response addressing the concerns and the corresponding changes in the manuscript.

Reviewer #1

Reviewer: In this work, a variety of large bandgap vdW materials were calculated to show their probable dielectric properties for potential usage as dielectric free of interface scattering in 2D electronics. Their out-of-plane dielectric constant, bandgap, electron affinity and leakage current were calculated, and 7 promising 2D dielectric materials were picked out. This work offered a significant guidance of the development of dielectrics with vdW surface for 2D electronics. Hence, the reviewer would recommend the publication in Nature Communications if the authors can address the following issues appropriately, especially the first one:

Author reply: We thank the reviewer for their favorable comments.

Reviewer: Main issue 1:

The benchmarking of these materials with the former reported ones cannot be carried out, as the evaluation criterion of the leakage current of the dielectrics must be unified:

The unit of leakage current density J is A/cm^2 intrinsically (and obtained directly), rather than A/cm or $pA/\mu m$. Please show the leakage current J count by A/cm^2 in the figures for a comparison with previous works. Note that the leakage current count by $pA/\mu m$ is strongly dependent on the device size and structure, but no details of the device model used was shown. Besides, please double check whether the unit is misused.

The leakage current should be calculated at the gate voltage of 1V other than 0.7V in order to make it comparable with previous works.

Author reply: We thank the reviewer for raising this point. In the previous version of the manuscript, we had calculated all currents assuming an 18 nm gate length, based on the 2020 IRDS [77], and this is how we arrived at the $pA/\mu m$ units. We apologize for not mentioning the gate length in the previous version of the manuscript. However, we agree that expressing in terms of A/cm^2 is more general. We appreciate that the reviewer suggested 1 V gate voltage instead of 0.7 V, but we respectfully kept 0.7 V gate voltage since we want to retain the comparison of the leakage current with the 2020 IRDS criteria [77]. Modifications to the manuscript were made as follows:

- In Fig. 4, we now plot the leakage current density with units of A/cm^2 . The shaded area shows where the leakage current exceeds the 2020 IRDS criteria [77]. The leakage criterion is given for an 18 nm gate with a 100 $pA/\mu m$, at a gate width of 107 nm per 28 nm device pitch, the acceptable current density through the gate is $0.145 A/cm^2$.

Reviewer: The effective mass, m^* , has great influence on J_{tunnel} , and differs a lot between materials. Effective mass can be extracted from the band structure of CBM and VBM, obtained alongside with the calculation of bandgap. If possible, make a list of the calculated m^* , and recalculate the leakage current with them.

Author reply: We had already given the effective mass some thought during our manuscript preparation and with the reviewer's encouragement we computed the out-of-plane effective masses of all materials from the bulk band structure. The results are shown in Table S4 and S5 in the Supplementary. We explained our methodology to extract the effective mass in the methods section and recalculated all the leakage currents for our 2D materials, shown in Fig. 4.

- In Section V.C, we added a sentence to the second paragraph and a paragraph to the methods section:

“We use a 0.7 V power supply voltage, V_{dd} , and 345 mV saturation voltage, V_{sat} , taken from International Roadmap for Devices and Systems (IRDS) [77] for setting an appropriate criterion for the leakage current and the calculation of electric field in the insulator. For the monolayers, we use the electron effective mass as the out-of-plane tunneling mass. For HfO_2 , we use a tunneling mass of $0.11m_e$ [87], hole effective mass of $0.58m_e$ [88, 89], band gap of 6 eV and electron affinity of 2.5 eV [90].”

“To extract the effective masses, we determine the location of the valence band maximum (holes) and conduction band minimum (electrons) along the high symmetry points of the first Brillouin zone, calculated using the PBE functional. The effective mass is extracted from the band curvature along the out-of-plane direction using

$$\frac{1}{m^*} = \frac{1}{\hbar^2} \frac{d^2E}{dk^2} \quad (5)$$

where $E(k)$ is the energy of the carrier at wavevector k , varying in the out-of-plane direction, and \hbar is the reduced Plank constant.”

- In Fig. 4, we now consider the effective out-of-plane electron/hole mass for each material instead of free electron mass, which was considered in the previous version. In response to a comment by Reviewer 3, we have also updated our estimates of the bilayer performance. The updated Fig. 4 and its caption are:

Fig. 4. Leakage current and EOT for monolayers (green circles), bilayers (red squares), and trilayers (grey triangles) in n-MOS applications. Reference dielectrics are also shown: The blue star shows monolayer h-BN. The dark purple line shows the EOT corresponding to the different thicknesses of HfO₂ (between 1-10 nm), while the orange line represents 0.4 nm of interfacial SiO₂ in addition to HfO₂. The shaded light grey area shows values that fall outside of the IRDS leakage current criteria (per pitch) criterion of 100 pA/μm for a transistor with a 28 nm pitch, an 18 nm long gate and an effective gate width of 107 nm [77]. The acceptable current density becomes 0.145 A/cm². For the bilayer, we estimate the performance using the dielectric constant of monolayer (filled squares) and using the bulk dielectric constant (hollow squares).

- Supplementary: We changed Fig. S3 and updated Table S4 and Table S5. We calculated and added the electron and hole effective masses of each material in Table S4 and S5, respectively.

Due to the change in the calculation of the leakage current, accounting for the out-of-plane effective mass, some previously identified materials (CaHI, SrBrF, SrHI) no longer meet the criteria that we put forward. Instead, three other materials, HoOI, SrI₂, and YOBr are added to the final list of promising candidates. We have amended our manuscript as follows:

- Section III: We updated paragraphs # 11, 12, 13 and 14:

“In Fig. 4, we show the performance of monolayer (1L), and bilayer materials, compared to HfO₂. Since calculation of multilayer (bilayer, trilayer, etc.) are computationally expensive we estimate their performance based on the dielectric constant of the monolayer (filled squares) and bulk (hollow squares) material, connected by a line. This representation shows the uncertainty in the dielectric response of the bilayer in our estimate. We show the calculated equivalent oxide thickness (EOT) with respect to the calculated leakage current for an n-MOS. Materials closer to the lower left are better gate dielectrics, featuring low EOTs with low leakage currents. We identify several monolayer and bilayer materials, which outperform HfO₂ with a 0.4 nm interfacial layer of SiO₂[78]. Moreover, we identify 8 monolayer dielectrics that outperform pure bulk HfO₂: HoOI, LaOBr, LaOCl, LaOI, SrHBr, SrI₂, TIF, and YOBr. These materials feature leakage current densities ranging from 10⁻⁵ A/cm² to 10⁻¹⁹ A/cm² and an EOT ranging from 0.05 nm to 0.5 nm.

Out of these materials, the most promising is monolayer LaOCl, having the lowest EOT (~0.05 nm), by a fair margin, among all materials, and with a leakage current less than 10⁻⁷ A/cm². Furthermore, even bilayer LaOBr outperforms bulk HfO₂ with an EOT < 0.5 nm and

leakage currents $< 10^{-18} \text{ A/cm}^2$ making these rare-earth oxyhalides the most promising in our list of 32 materials. Note that while LaOBr is in the Materials Cloud “layered materials” database, LaOCl is not and it is a material we added from the Materials Project database. We limited our search to 32 materials so other promising rare-earth oxyhalides not included in our present investigation, like GdOCl or YOCl, may also be promising gate dielectrics to be identified in future investigations”.

“Recently, two new dielectrics have been proposed for vdW materials, CaF_2 and Bi_2SeO_5 . CaF_2 has been shown to have a desirable dielectric constant of 8.4, and an enormous band gap of 12.1 eV [36]. The thermally stable Bi_2SeO_5 has been demonstrated with a dielectric constant of 21 and a moderate band gap of 3.9 eV [37]. However, while both CaF_2 and Bi_2SeO_5 outperform other bulk dielectrics such as HfO_2 , CaF_2 is not a vdW material and is prone to the same surface roughness and interface defects of conventional oxides. It thus remarkable that we have identified 8 monolayer vdW materials that outperform HfO_2 , the industry-leading bulk dielectric, without even considering the intrinsic benefits of vdW dielectrics, e.g., perfect interfaces without defects. All 8 monolayers exhibit high band gaps ($> 3 \text{ eV}$), high dielectric constants (> 5.8), tiny leakage current ($< 10^{-5} \text{ A/cm}^2$), small EOT ($< 0.6 \text{ nm}$) and suitable band offsets”.

“Our most promising materials LaOBr and LaOCl, which outperform HfO_2 , CaF_2 , and Bi_2SeO_5 , are known stable and readily available materials. LaOBr and LaOCl are water insoluble and have been investigated for applications as scintillators and ion transport channels [51, 53, 54]. Previously, LaOBr and LaOCl have been synthesized using a solid-state reaction between Lanthanum Oxide (La_2O_3) and ammonium chloride/ammonium bromide ($\text{NH}_4\text{Cl}/\text{NH}_4\text{Br}$). We could not find any literature on attempts to exfoliate or characterize monolayers of LaOCl or LaOBr. Our calculations show that they are not just layered but in fact exfoliable and show that LaOBr and LaOCl have the potential to realize highly performant true vdW field-effect transistors. We hope that our result encourages further experimental investigation into the materials we identified (HoOI , LaOBr, LaOCl, LaOI, SrHBr , SrI_2 , TlF, and YOBr) and specifically into the monolayer and bilayer form of the rare-earth oxyhalides LaOBr and LaOCl”.

- We also slightly revised paragraphs # 3 and 4 in section III:

“Inspecting **Table 1** reveals that, in general, the optical dielectric constant is significantly lower than the corresponding static dielectric constant, indicating a large ionic contribution to the dielectric response for all materials under consideration. Considering the out-of-plane direction, the optical dielectric constant ($\epsilon_{\infty, \perp}$) ranges from 2.4 (SnF_4) to 4.9 (BiOCl) for bulk, and from 2.8 (MgCl_2) to 5.8 (CaHI) for monolayers. In contrast, the static dielectric constant ($\epsilon_{0, \perp}$) is as high as 28.2 for bulk PbClF and 55.8 for monolayer LaOCl. In the supplementary (Table S6) we list the experimentally determined values of the dielectric constant for CdBr_2 , CdCl_2 , and PbI_2 and find agreement within 20% between the theoretical and experimental values [65, 66, 67].”

“Finally, note that compared to their bulk forms, monolayer LaOBr, LaOCl, SrBrF , and SrI_2 are unique in showing a significantly improved out-of-plane electronic dielectric response, while they do not see a significant change in their ionic response. Of these materials, LaOBr has the third highest static out-of-plane dielectric constant among all monolayers (13.2). Only LaOCl (55.8) and PbClF (15.2) have higher out-of-plane dielectric constants, although the ionic dielectric response in monolayer PbClF is significantly reduced compared to bulk, while it is strongly enhanced in monolayer LaOCl. Based only on their out-of-plane dielectric constants, LaOBr, LaOCl, and PbClF could be good candidates for a gate dielectric, if they turn out to be good insulators as well.”

- Section IV: We updated our conclusion section based on our new finding:

“Starting from a database of layered materials, we selected 32 viable candidates for suitable vdW dielectric applications (exfoliable, good band gap, and stable). For each material, we calculated the in-plane and out-of-plane macroscopic dielectric constants using first principles. Our calculations show a wide range of in-plane and out-of-plane dielectric values, from 2.56 to 98.37. To gauge the performance of each material as a gate dielectric in nMOS applications, we calculated the leakage current and the EOT for each material. Since Hydrobromides have been reported as being hygroscopic, we exclude SrHBr from the shortlist of candidate and ended up with seven promising vdW dielectrics: HoOI, LaOBr, LaOCl, LaOI, SrI₂, TlF, and YOBr. The best performing material, monolayer LaOCl, shows immense promise as a gate dielectric, with an EOT < 0.1 nm while maintaining leakage currents < 10⁻⁷ A/cm². Furthermore, LaOBr and LaOCl are known and stable materials. We hope that our research leads to the further exploration of rare-earth oxychlorides and oxybromides for applications as layered dielectrics.”

Reviewer: These two articles for the evaluation criterion of the leakage current of gate dielectrics: (John Robertson & Robert M. Wallace, *Materials Science and Engineering R*, 2015, 88, 1–41, especially Fig.2 and Fig.9; Yury Yu. Illarionov et al. *Nature Communications*, 2020, 11,3385, especially Fig.3). These references repay careful reading.

Author reply: We thank you the reviewer for suggesting these two outstanding articles. We have added the following articles to the reference section of our manuscript:

[69] Robertson, John, and Robert M. Wallace. "High-K materials and metal gates for CMOS applications." *Materials Science and Engineering: R: Reports* 88 (2015): 1-41.

[74] Illarionov, Yury Yu, et al. "Insulators for 2D nanoelectronics: the gap to bridge." *Nature Communications* 11.1 (2020): 1-15.

Reviewer: Other issues:

1. Article Ref. 32 was cited one-sidedly in page 2. The article offered the method to avoid the problem of non-uniformity with a PTCDA buffer layer. The author can find its shortages such as a thicker EOT, but should not miss the information it brought about.

Author reply: We thank the reviewer for their clarification.

- Section I: We updated the third paragraph of the introduction (page 2):

“However, the selection of gate dielectrics to use for vdW materials has not received as much attention. Most TMD-based MOSFETs investigated to date use atomic-layer deposited (ALD) oxides like HfO₂ and Al₂O₃ [26, 27, 28, 29]. Unfortunately, when an ALD oxide is deposited on a 2D material, the naturally terminated surfaces now become a large drawback because covalent bonds between the oxide and the 2D material are hard to make [30, 31]. Non-uniform nucleation will give rise to a non-uniform thickness and, in the absence of a uniform thin dielectric, MOSFET performance will become unacceptably poor. One proposed solution addressing the non-uniformity is the deposition a perylene-tetracarboxylic dianhydride (PTCDA) molecular crystal layer [32]. Moreover, where covalent bonds are formed, the natural 2D material termination is broken and the surface states we wanted to avoid are reintroduced. It is thus hard to foresee how ALD oxides can ever be a component of a 2D material MOSFET technology.”

Reviewer: 2. Some of the materials (especially in cat. 2) are common and their dielectric constant (experimental data) can be found. Please display them in the same chart to reflect the error range of the calculation.

Author reply: We found three experimental papers for the bulk forms of PbI₂ and CdBr₂, and CdCl₂ in which their dielectric constants are reported. We have amended our Supplementary as follows:

- The following references are added to the supplementary:

[S1] R. D. Bringans and W. Y. Liang, "The dielectric functions of CdI₂, CdBr₂ and CdCl₂," *Physica B+C*, vol. 99, pp. 276-280, 1980.

[S2] D. P. Yadav, K. V. Rao and H. N. Acharya, "Dielectric properties of PbI₂ single crystals.," *physica status solidi (a)*, vol. 60.1, pp. 273-276, 1980.

[S3] S. Ekhard, B. Palosz and B. Wruck., "In situ observation of the polytypic phase transition 2H-12R in PbI₂: investigations of the thermodynamic structural and dielectric properties," *Journal of Physics C: Solid State Physics*, vol. 20.26, p. 4077, 1987.

- Table S6 in the supplementary compares the static dielectric constants of our calculation with the experimental values obtained from the aforementioned references:

Table S6. Our calculation for the static dielectric constants of CdBr₂, CdCl₂, and PbI₂ versus the available experimental dielectric constants [S1, S2, S3]. The frequencies at which dielectric constants are obtained are mentioned for each material. The experimental values are measured from different methods.

Experimental Data	Material	Bulk (ϵ_{∞})	Bulk (ϵ_0)
		\perp	\perp
Cat. 2	CdBr ₂	3.9 [S1] (10 ¹⁴ Hz)	-
	CdCl ₂	3.0 [S1] (10 ¹⁴ Hz)	-
	PbI ₂	-	6.7 [S2, S3] (10 ⁶ Hz)

Comparing with the calculated values, Ref. 1 reports a higher measured dielectric constant (3.9 exp vs 3.6 theory) for CdBr₂ and a lower dielectric constant for CdCl₂ (3.0 exp vs 3.7 theory) while the PbI₂ dielectric constant is a little higher as well (6.7 exp vs 6.1 theory). Note also that the experimental references were not necessarily conducted with the goal of accurately determining the low-frequency dielectric constant.

- We added the following sentence to the Section III of our manuscript:

"In the supplementary (Table S6) we list the experimentally determined values of the dielectric constant for CdBr₂, CdCl₂, and PbI₂ and find agreement within 20% between the theoretical and experimental values [65, 66, 67]."

Reviewer: 3. In this work, LaOCl was found having an ultrahigh out-of-plane dielectric constant (55.8), far more higher than its bulk (11.8). To ensure the reliability, the dielectric constant of few-layer LaOCl should be provided in supporting information.

Author reply: We calculated the dielectric constant of bilayer LaOCl and reported in Table S7.

In response to reviewer 3, we also reported the dielectric constant of monolayer LaOCl with three different vacuum sizes in Table S7. Unfortunately, while the electronic response is well converged, the ionic out-of-plane response does show a significant sensitivity to the size of vacuum used in the DFPT calculation and is not converged even at the strictest energy threshold. We attribute this to the limitations in the accuracy of the DFPT in VASP. The latter is not entirely surprising as our dielectric constant rescaling procedure, that removes the vacuum contribution, amplifies errors, and phonon-related quantities, especially those involving polarization vectors are known to have much larger errors compared to the ground state energy. For LaOBr the results do not show the same errors, and neither are similar errors expected for other materials with lower ionic dielectric responses. In the main manuscript, we report all values obtained for a 15 Å vacuum.

- In Section V.B.3, we added one more paragraph to the method as follows:

“We also studied the sensitivity of our method with respect to vacuum size. While our results are converged for most materials, we found that, for materials with extremely large out-of-plane dielectric constants, (e.g. LaOCl) significant errors are observed (see Table S7 of the Supplementary). Unfortunately, the ionic out-of-plane response does show sensitivity to the vacuum used and is not converged even at the strictest energy threshold. We attribute this to the limitations in accuracy of the DFPT in VASP. The latter is not entirely surprising as the dielectric constant rescaling procedure amplifies errors, and phonon-related quantities, especially those involving polarization vectors, are known to have much larger errors compared to the ground state [86].”

- We updated the Reference section by adding this article:

*[86] Gaddemane, Gautam, et al. "Limitations of ab initio methods to predict the electronic-transport properties of two-dimensional semiconductors: the computational example of 2H-phase transition metal dichalcogenides." *Journal of Computational Electronics* 20.1 (2021): 49-59.*

- In the Supplementary, we added Table S7:

Table S7. The dielectric constant of monolayer, under four different vacuum conditions, and bilayer LaOCl.

		Vacuum (Å)	ϵ_{∞}		ϵ_0	
			\perp	\parallel	\perp	\parallel
LaOBr	Monolayer	15	5.32	4.66	13.21	18.24
		25	5.32	4.67	13.09	18.38
		30	5.32	4.67	13.18	18.33
		35	5.32	4.66	12.93	18.39
LaOCl	Monolayer	15	5.53	4.54	55.80	21.28
		25	5.53	4.54	67.62	22.85
		30	5.54	4.54	44.05	21.46
		35	5.53	4.54	80.68	23.73
	Bilayer	32	4.81	4.50	12.52	21.43

Reviewer: 4. In page 2, “Nevertheless, neither CaF₂ nor Bi₂SeO₅ are layered compounds with unterminated bonds at the surface, raising questions of passivation to eliminate interface states and ensure reliability”. Actually, Bi₂SeO₅ is a layered material without dangling bonds between the gap. The author

needs to correct related statements. Besides, single-crystalline, polycrystalline and amorphous Bi₂SeO₅ should be distinguished.

Author reply: We thank the reviewer for bringing this point to our attention. We have calculated the exfoliation energy of Bi₂SeO₅ as 36.52 meV/Å², indicating it is potentially exfoliable. Nevertheless, our materials exfoliation energies are lower than the Bi₂SeO₅.

- We highlighted the revised sentence in paragraph #4 of Introduction as follow:

“h-BN is a vdW material that has successfully been used as a dielectric in transistors, in combination with vdW channel materials [6, 33]. However, h-BN also has significant drawbacks such as a low dielectric constant, which is undesired, and the requirement to either transfer h-BN or to grow at high temperatures that are not compatible with semiconductor technology [34, 35]. Recently, the crystalline dielectric CaF₂ has also been investigated to avoid the drawbacks of amorphous oxides [36, 37]. Nevertheless, unlike Bi₂SeO₅, which is a layered material, CaF₂ is not a layered compound with unterminated bonds at the surface, raising questions of passivation to eliminate interface states and ensure reliability. Bi₂SeO₅ does present an interesting native layered oxide although the thickness of a single layer Bi₂SeO₅ (~11.47 Å) is significantly thicker than most 2D materials.”

- In Section III, we amended our text:

“Recently, two new dielectrics have been proposed for vdW materials, CaF₂ and Bi₂SeO₅. CaF₂ has been shown to have a desirable dielectric constant of 8.43, and an enormous band gap of 12.1 eV [36]. The thermally stable Bi₂SeO₅ has been demonstrated with a dielectric constant of 21 and a moderate band gap of 3.9 eV [37]. However, while both CaF₂ and Bi₂SeO₅ outperform other bulk dielectrics such as HfO₂, CaF₂ is not a vdW material and is prone to the same surface roughness and interface defects of conventional oxides. It thus remarkable that we have identified 8 monolayer vdW materials that outperform HfO₂, the industry-leading bulk dielectric, without even considering the intrinsic benefits of vdW dielectrics, e.g., perfect interfaces without defects. All 8 monolayers exhibit high band gaps (> 3 eV), high dielectric constants (> 5.8), tiny leakage current (< 10⁻⁵ A/cm²), small EOT (< 0.6 nm) and suitable band offsets”.

Reviewer: 5. There are some minor errors that may be clerical errors. “either transfer h-BN or to grow at temperatures” should be “either transfer h-BN or to grow at high temperatures”. (page2, line18) The unit of exfoliation energies should be “meV/Å²” instead of “eV/Å²”. (page4, line14-17)

Author reply: We thank the reviewer for pointing out the typo.

- Section I: We added word “high” to the fourth paragraph of the introduction.

“h-BN is a vdW material that has successfully been used as a dielectric in transistors, in combination with vdW channel materials [6, 33]. However, h-BN also has significant drawbacks such as a low dielectric constant, which is undesired, and the requirement to either transfer h-BN or to grow at high temperatures that are not compatible with semiconductor technology [34, 35]. Recently, the crystalline dielectric CaF₂ has also been investigated to avoid the drawbacks of amorphous oxides [36, 37] Nevertheless, unlike Bi₂SeO₅, which is a layered material, CaF₂ is not a layered compound with unterminated bonds at the surface, raising questions of passivation to eliminate interface states and ensure reliability. Bi₂SeO₅ does present an interesting native layered oxide although the thickness of a single layer Bi₂SeO₅ (~ 11.47 Å) is significantly thicker than most 2D materials.”

- Section III: We corrected the unit of exfoliation energy in the first paragraph.

“To ensure all materials are in fact layered, we first calculate the exfoliation energies, E_{ex} . The exfoliation energy ranges from 3.12 meV/Å² PbI₂ (Cat.2) to 40.22 meV/Å² LaOCl (Cat.1b) and the values for each material are listed in the Supplementary Information (Supplementary Fig. S1). As a rule-of-thumb, materials with $E_{ex} < 100$ meV/Å² are considered easily exfoliable compounds [38]. Using this criterion, all materials under consideration are layered and exfoliable.”

Reviewer: 6. The chemical instability of H- is well-known. Thus, it's better not to recommend the ones with H- in the conclusion.

Author reply: We could not find much information on the stability of the hydrobromides and hydro-iodides but agree that common sense would indicate that they are unlikely to be stable materials.

- Section IV: We have revised the conclusion section:

“Starting from a database of layered materials, we selected 32 viable candidates for suitable vdW dielectric applications (exfoliable, good band gap, and stable). For each material, we calculated the in-plane and out-of-plane macroscopic dielectric constants using first principles. Our calculations show a wide range of in-plane and out-of-plane dielectric values, from 2.56 to 98.37. To gauge the performance of each material as a gate dielectric in nMOS applications, we calculated the leakage current and the EOT for each material. Since hydrobromides are generally hygroscopic materials, we exclude SrHBr from the shortlist of candidate and ended up with seven promising vdW dielectrics: HoOI, LaOBr, LaOCl, LaOI, SrI₂, TlF, and YOBr, all of which promise better performance than HfO₂. The best performing material, monolayer LaOCl, shows immense promise as a gate dielectric, with an EOT < 0.1 nm while maintaining leakage currents < 10⁻⁷ A/cm². Furthermore, LaOBr and LaOCl are known and stable materials. We hope that our research leads to the further exploration of rare-earth oxychlorides and oxybromides for applications as layered dielectrics.”

Reviewer #2

Reviewer: In this manuscript, the authors analyze a database of 2D materials and refine the existing calculations in order to find a subset of promising materials as gate dielectrics. The authors use more accurate calculations to improve the estimate of the electronic properties, and analyze criteria such as dielectric constants, band gaps and leakage currents. The authors identify some 2D materials which could be useful for 2D devices. The results are interesting and I recommend publication after minor revisions.

Author reply: We sincerely thank the reviewer for their favorable comments.

Reviewer: * At the end of page 3, the authors state "After imposing the >2.5 eV PBE bandgap criterion, pruning the aforementioned materials and adding some similar compounds discovered on AFLOW and Materials Project, ..."

The selection from the Materials Cloud dataset is described clearly. However, the authors should specify what "similar compounds" means and how they have been found. Especially since one of the proposed materials LaOCl is picked from these two databases.

Author reply: We thank the reviewer for noticing this point. Ultimately, we had to limit the materials dataset and the choice of 32 materials can perhaps be considered an “arbitrary” cut-off. For materials on Materials Cloud dataset, such as LaOBr, we have just arbitrarily picked another from the Materials Project database, such as LaOCl, which the same chemical formula and structure. At the end of page 3, the phrase of “similar compounds” refers to materials with the same structure (*i.e.* space group, crystal system).

- We have updated paragraph #6 in section II (Materials selection) of the manuscripts as below:

“Before proceeding with calculations, we prune the dataset through manual inspection. We exclude LiBH_4 since it is a deliquescent solid-state material (melting point of $275\text{ }^\circ\text{C}$) at ambient conditions and is highly sensitive to water and oxygen [45]. We also remove NaCN because of its toxic and corrosive properties and its danger to the environment. RbCl is another material we remove from our dataset since, while Materials Cloud identifies it as a potential 2D material, the bulk does not present a layered structure. LiOH , NaOH , $\text{Mg}(\text{OH})_2$, and $\text{Ca}(\text{OH})_2$ are also eliminated since they are elemental bases which are very soluble in water and invariably appear as water complexes. We also did not recalculate the values for $h\text{-BN}$ as the dielectric properties have been reported accurately and in detail previously [34]. After imposing the $>2.5\text{ eV}$ PBE bandgap criterion, pruning the aforementioned materials and adding some similar compounds (i.e. space group, crystal system) discovered on AFLOW and Materials Project, we obtain a list containing 32 vdW materials, all of which are halides”.

- Also, the caption of Fig. 1 in the manuscript is changed to:

“Fig. 1. A) Side view of the monolayer structures. B) Top view of the monolayer structures, where the yellow squares represent the computational unit cells. C) Side view of the bilayer structures, showing the A-A and A-B stacking configurations. The measurement of the thickness of the monolayers (t) is indicated. Category 1 contains three similar structures, further divided in subcategories a, b and c.”

Reviewer: * Figure S1 (probably, I can't read the caption) is truncated at the bottom.

Author reply: We apologize for the inconvenience. We have corrected Fig. S1 in the Supplementary. We have copied Fig. S1 here as well:

Fig. S1. The exfoliation energies (E_{ex}) obtained from PBE-DFT. The value of $100\text{ meV}/\text{\AA}^2$ is considered as the E_{ex} threshold if a material is an easily or potentially exfoliable. The green dot and the red dot respectively belong to the PbI_2 (easily exfoliable) and LaOCl (potentially exfoliable).

Reviewer: * The in-plane dielectric constants are found as an average of the two cartesian directions ($\epsilon_x + \epsilon_y$)/2. However, are the materials isotropic? If not, what is the degree of anisotropy? This consideration is likely important for manufacturing devices.

Author reply: We thank the reviewer for raising this point. Only 3 materials (AlOCl, InOCl, and ScOBr) out of 32 materials in our list show anisotropic behavior ($\epsilon_x \neq \epsilon_y$). The dielectric constants of the aforementioned materials with anisotropic behavior are now added to the Table S8 of the supplementary:

Table S8. Dielectric constant values for anisotropic materials.

	Material	Bulk (ϵ_∞)		Bulk (ϵ_0)		Monolayer (ϵ_∞)		Monolayer (ϵ_0)		
		x	y	x	y	x	y	x	y	
Cat. 1	c	AlOCl	3.3	2.8	10.5	5.1	3.3	2.8	10.5	5.1
		InOCl	4.1	3.6	10.3	7.6	4.0	3.5	10.2	7.5
		ScOBr	4.7	3.7	14.4	9.4	4.7	3.7	14.3	9.3

Reviewer #3

Reviewer: In the manuscript “Identification of Two-Dimensional Layered Dielectrics from First Principles” Osanloo et al. present a systematic first principles study of novel layered materials with a focus on their potential as gate dielectrics as needed for field-effect transistor devices. The authors extract 32 promising candidates from material databases and calculate their dielectric properties (in bulk and monolayer), electron affinities, and band gaps. From this, they identify eight candidates which outperform HfO₂.

The presented study is interesting and the manuscript is well written and certainly merits publication, but I’m afraid that it is in its present form not suitable for the broad readership of Nature Communications due to two main reasons:

Author reply: We sincerely thank the reviewer for their time and comments on our manuscript and hope our answers will satisfy the reviewer’s concerns.

Reviewer: 1.) The manuscript is not very accessible for non-expert readers who are not familiar with the functional basics of field-effect transistor devices and their realizations using layered materials. As an example the abstract and the introduction (more or less) directly start with the discussion of “channel” and “gate” materials without explaining their individual roles, while the title is rather general. Also, the current role of HfO₂ is not entirely clarified so that the reoccurring comparison to it lacks context.

Author reply: We thank the reviewer for their comments which will help the paper appeal to the readers of Nature Communication journal. We have updated our manuscript as follows:

- We have made our abstract more general:

“The physical limit in the size reduction of electronic devices considered as a huge obstacle in the integration of three-dimensional (3D) dielectric materials into electronic and optoelectronic devices. Moreover, most conventional 3D dielectrics suffer from relatively low dielectric constants that limit their performance in managing charge carrier flow in the channels. In this regard, two-dimensional (2D) van der Waals (vdW) materials simultaneously promise device dimension-

reduction and ideal electrostatic control of charge carrier flow in a channel free of surface roughness or defects. To realize this ideal, good vdW dielectrics are needed in addition to the well explored channel materials. We study the dielectric properties of 32 easily exfoliable vdW materials using first principles methods. Specifically, we calculate the static and optical dielectric response of the monolayer and bulk form. In monolayers, we discover a strong out-of-plane response in GeClF (11.0), LaOBr (13.2), LaOCl (55.8) and PbClF (15.2), while the in-plane dielectric response is strong in BiOCl, PbClF, and TlF, ranging from 64.9 to 98.4. To assess their potential as gate dielectrics, we calculate the bandgap and electron affinity, and estimate the leakage current through the dielectric. **We discover seven monolayer 2D dielectrics that promise to outperform bulk HfO₂: HoOI, LaOBr, LaOCl, LaOI, SrI₂, TlF, and YOBr** with lower leakage currents at a significantly reduced equivalent oxide thickness. Of these, LaOBr and LaOCl are the most promising and our findings motivate the growth and exfoliation of rare-earth oxyhalides for their use as vdW dielectrics on vdW transistor channel materials.”

- We have added a few sentences to the third paragraph of Section I to make our message clearer:
“Suitable gate dielectric materials are a critical component that allows the “gate” to exercise electrostatic control of the “channel” where electrons flow. The dielectric both blocks current flow between the gate and channel (gate leakage) and enhances the electrostatic displacement field (electrostatic control). However, the selection of gate dielectrics to use for vdW materials has not received as much attention. Most TMD-based MOSFETs investigated to date use atomic-layer deposited (ALD) oxides like HfO₂ and Al₂O₃ [26, 27, 28, 29].”

Reviewer: 2.) There are a variety of methodological questions open (see below). Without the corresponding clarifications it is not clear how reliable the ab initio results are. The following first lists the open methodological questions followed by a list of minor remarks.

Author reply: We thank the reviewer for his/her comments. The methods we use are state-of-the-art and give valuable insights at this point in time, nevertheless we would hope that future developments in ab initio methods can further improve on methodologies and make future predictions more accurate.

Reviewer:

A) It is well known that due to reduced “environmental screening” in layered materials (in the bulk and especially in the monolayer) these materials hosts enhanced long-range Coulomb interactions, which decisively affect (enhance) the band gaps of layered semi-conductors. These effects are not captured in conventional LDA or GGA based DFT calculations and can only be properly taken into account within many-body perturbation theory in form of GW calculations (see e.g. Phys. Rev. Lett. 111, 216805, 2013 or Phys. Rev. B 94, 155406, 2016). As an approximation HSE functionals might be used to enhance the LDA/GGA band gaps, but it is a priori not clear how well HSE performs for specific layered materials and whether the same amount of exact exchange inter-mixing is needed for the monolayer and bulk systems. Since the authors used here the HSE functional for the monolayer band-gap calculations it is not clear how reliable these values are. I would suggest to perform at least for one candidate from each category a G₀W₀ calculation to check and define the HSE accuracy for the specific monolayers.

Author reply: We thank the reviewer for this point.

Before proceeding, we must clarify here that the HSE06 calculations are employed only to calculate the material’s affinity and bandgap, not the dielectric constants or interlayer distance. We agree that the physics taken into account by the GW method is superior to that of the HSE06 hybrid functional approach. But for the purpose of computing the affinity and band bandgap, and since we are considering 32 materials, we feel that the use of the HSE method is the most appropriated choice balancing computational expense while avoiding the underestimation of the bandgap that occurs with LDA or GGA.

Second, the computation of the ionic component of the dielectric constant requires Density-Functional Perturbation Theory (DFPT). Unfortunately, DFPT using HSE (or GW) is currently not implemented in VASP.

Concerning, the reviewers' comments about the long-range Coulomb interactions: Fortunately, we only report on the static long-wavelength ($q \rightarrow 0$) dielectric constant. This strongly reduces the severity of the long-range interactions since the Coulomb kernel of a 1-dimensional dipole is a step function and does not have a long-range interaction unlike a 2- or 3-dimensional dipole. To show that the long-range interactions are not an issue, we performed the monolayer calculations for LaOCl and LaOBr with three additional different vacuum sizes (see Table S7). The extracted dielectric constants do not change significantly except for the ionic response in LaOCl where the error propagation in the DFPT calculation affects the results, as explained in our response to Reviewer 1. Nevertheless, these results indicate that environmental screening by periodic images do not affect the obtained dielectric constants.

Accordingly, we changed the following sections of the manuscript as follows:

- To Section III, we added the following paragraph #7:

“Although there are superior methods available (e.g., GW) compared to hybrid functionals, they come at a significant computational cost. HSE balances computational expense while avoiding the underestimation of the bandgap resulting from LDA or GGA. In this work, we only report on the long-wavelength dielectric constant [73]. This strongly reduces the severity of the long-range interactions that may adversely affect supercell methods. Since the Coulomb kernel of a 1-dimensional dipole is a step function, it does not have a long-range interaction unlike a 2- or 3-dimensional dipole. To show that the long-range interactions are not an issue, we performed the monolayer calculations for LaOCl and LaOBr with three additional different vacuum sizes (see Table S7 in the Supplementary). The extracted dielectric constants do not change significantly except for the ionic response in LaOCl where the error propagation in the DFPT calculation affects the results. Nevertheless, these results indicate that environmental screening by periodic images does not affect the obtained dielectric constants.”

- The below paper is added to the References:

*[73] Rasmussen, Filip A., et al. "Efficient many-body calculations for two-dimensional materials using exact limits for the screened potential: Band gaps of MoS₂, h-BN, and phosphorene." *Physical Review B* 94.15 (2016): 155406.*

Reviewer: B) The authors study the ability to exfoliate monolayers for each material by calculating the exfoliation energies from bulk and monolayer calculations. I see two problems with this:

B-1) Again it is not clear to me how reliable these exfoliation energies as gained from GGA+Grimme correction are in comparison to full RPA calculations.

Author reply: We appreciate reviewer's concern regarding the exfoliation energy calculation. It is shown in Refs. [83, 84], that nonlocal correlation functional methods (NLCF), that account for e.g. nonlocal van der Waals (vdW) interactions, are in satisfactory agreement with experimental data for a wide variety of materials. Although GGA+Grimme is anticipated to be less accurate than RPA in the long-range limit, its performance has been shown to be accurate for the short-range interactions involved in layered solids.

A comprehensive study on the performance of RPA, PBE-D, LDA, vdW-DF1 and vdW-DF2 has been added as new Refs. [83,84] in the Reference section:

*[83] Ashton, Michael, et al. "Topology-scaling identification of layered solids and stable exfoliated 2D materials." *Physical review letters* 118.10 (2017): 106101.*

*[84] Björkman, Torbjörn, et al. "van der Waals bonding in layered compounds from advanced density-functional first-principles calculations." *Physical review letters* 108.23 (2012): 235502.*

Reviewer: B-2) The ability to exfoliate monolayers from bulk vdW materials should to my mind not just be defined by the exfoliation energy but also by the monolayer stability. The latter can, for example, be studied by calculating the phonon dispersions from DFPT. This should always be checked when novel monolayer materials are predicted and investigated.

Author reply: We thank the reviewer for his/her comment. We report the energy of the phonon modes at Gamma using DFPT for the monolayer and bulk of all materials and reported the data in Table S9 and Table S10 in supplementary. We consider a full phonon spectrum beyond the scope of the current paper, especially since no free-standing materials are envisioned as a final application. We have also addressed this point in our manuscript.

- Section III: We added the following paragraph in the first paragraph of the results section:

“Moreover, we calculate the phonon energies of monolayer and bulk for all materials to investigate the stability of our monolayers. We list the value of monolayer and bulk form phonon energy in Tables S9 and S10 of the supplementary, respectively. The phonon energy calculation shows that all monolayer materials, except TIF and GeClF are predicted to be stable.”

- Section V.B.2: We added a subsection named “Phonon calculation” in which we explain the phonon energy calculation for both monolayer and bulk of all materials. The following paragraph is added:

“We calculate the phonon energies and vibrational modes for both monolayer and bulk of all materials from DFPT. Phonon energy calculations, acoustic phonons in particular, indicates the stability of a system. We report the monolayer and the bulk Phonon energies in Tables S9 and Table S10 of the supplementary, respectively.”

- Supplementary: We added Table S9 and Table S10 reporting the values of phonon energies for the monolayer and bulk of all materials.

Reviewer: C) Based on the comment from B-2) I also wonder how strong the in-plane ϵ_0 of TIF, PbClF, and BiOCl in bulk and monolayer might be influenced by (very) soft phonons?

Author reply: We have not observed any correlation between the soft phonons and the in-plane dielectric constants. Although, in some materials such as LaOBr and LaOCl the impact of soft phonons on the in-plane ϵ_0 is negligible, there are some other materials such as TIF that we clearly observe a strong impact of soft phonons on the in-plane ϵ_0 . Please find Table S9 and Table S10 in the Supplementary where we report the phonon dispersion energies for monolayer and bulk of all materials.

Reviewer: D) It looks like that for the calculation of the dielectric properties the PBE+Grimme exchange functional has been used instead of the HSE one as used for the bandgap calculations in the monolayer limits. Since the value of the band-gap affects also the dielectric properties of a semiconductor the authors should either compare the results between PBE and HSE or should at least comment on this point.

Author reply: As we mentioned in our response to the first point raised, HSE DFPT is unfortunately not implemented in VASP at this time. However, while it is often simplistically claimed that “ $\epsilon \sim 1/E_g$ ”, this is generally known to be very crude and clearly not true from our results. Moreover, the dielectric response is determined by polarization, which is in turn determined by charge density, and while the band gap shows large differences between HSE and PBE+Grimme, the charge density shows a much smaller difference. Finally, we found reasonable agreement with experiments and the values we calculated.

Reviewer: C) The accuracy of the “Vacuum elimination [in the dielectric properties] from the 2D structures” is not entirely clear to me. From a conventional RPA point of view, in which the dielectric function is defined by $\epsilon = 1 - V \cdot \chi$ with V being the bare Coulomb interaction and χ the bare susceptibility, it is clear that the long-range V is responsible for spurious interlayer screening even for large supercells (χ is rather localized), which needs to be corrected for. For GW-like calculations this has been

discussed for example in Phys. Rev. B 94, 155406, 2016 or Phys. Rev. B 92, 2015. Here and in Ref. 34, the authors use DFPT based on PBE to calculate ϵ . Thus, I wonder if the applied “vacuum elimination” solves the same problems here as indicated from the RPA example above.

Author reply: We thank the reviewer for highlighting two papers. Unfortunately, the reference for the second paper was not complete and we were not able to access this paper. To address this concern, we performed the monolayer calculations for LaOCl and LaOBr with three additional different vacuum sizes (see Table S7 in the Supplementary). The extracted dielectric constants do not change significantly except for the ionic response in LaOCl where the error propagation in the DFPT calculation affects the results.

Reviewer: Minor points:

I) What is meant with “back-end” and “front-end” on page 2?

Author reply: We apologize for the confusing phrasing in the paper. The manufacturing process of an integrated circuit (IC) composed of several steps. Processing starts at the “front-end of line”, where the actual individual active devices (e.g., transistors, memory cells) are created in the silicon wafer. In the “back end of line” the devices are connected with each other to form logic circuits through various layers of metals that form the interconnects. Recently, research focusses on adding active components to the otherwise passive “back end” metallization, e.g., to save energy by turning off entire processing cores in mobile applications.

We agree that this is specialized terminology and we have updated our manuscript to make it more accessible to a wider audience. In Section I: We updated the second paragraph of our introduction.

“The field of nanoelectronics naturally invites the use of 2D materials since a reduction of the channel length and, more recently, channel thickness has been a driver for dramatic technological progress [6,7,8]. Moreover, surface states have limited the performance and reliability in many nano-electronic applications [9, 10, 11, 12, 13] whereas the naturally passivated surfaces of vdW materials alleviate the concern of surface states. As a result, TMDs are now actively being considered as channel materials by the semiconductor industry [15, 16]. High mobilities are reported [3, 16], doping techniques [17, 18] are under development, metal-oxide-semiconductor field-effect transistors (MOSFETs) are being fabricated [19, 20, 21], and contact technology is under investigation [22, 23]. TMDs are thus well on the way to commercial application in transistors, being investigated as a replacement of silicon in the active switching devices (“front-end”) of semiconductor technology but also as an augmenting technology in the metallization layers that interconnect the devices in the “back-end” [24, 25].”

Reviewer: II) It is not clear on which ab initio level the mono-, bi-, and tri-layer data shown in Fig. 4 has been calculated.

Author reply: We thank the reviewer for touching on this point. We only calculate the monolayer and bulk dielectric constants using the DFPT procedure outlined in the paper. The bilayer and trilayer were estimated from the monolayer results. However, in general, the dielectric constant in a bilayer is different than the monolayer dielectric constants. In our updated manuscript, we have improved our estimate by calculating the leakage and EOT for bilayer at two extremes: once with the dielectric constant of the monolayer and once with the dielectric constant of bulk. We amended our manuscript as follows:

- In Fig. 4, we removed the leakage current corresponds to the bilayers and trilayers; instead, we show the two extreme cases, connected by a line: filled squares represent the bilayer with the monolayer dielectric constant, while the hollow squares show the bilayers with the bulk dielectric constant. This representation clearly shows the uncertainty in our extrapolation of the bilayer performance.
- In Section III, we have updated the text accordingly:

“In Fig. 4, we show the performance of monolayer (1L), and bilayer materials, compared to HfO_2 . Since calculation of multilayer (bilayer, trilayer, etc.) are computationally expensive we estimate their performance based on the dielectric constant of the monolayer (filled squares) and bulk (hollow squares) material, connected by a line. This representation shows the uncertainty in the dielectric response of the bilayer in our estimate. We show the calculated equivalent oxide thickness (EOT) with respect to the calculated leakage current for an n-MOS. Materials closer to the lower left are better gate dielectrics, featuring low EOTs with low leakage currents. We identify several monolayer and bilayer materials, which outperform HfO_2 with a 0.4 nm interfacial layer of SiO_2 [78]. Moreover, we identify 8 monolayer dielectrics that outperform pure bulk HfO_2 : HoOI , LaOBr , LaOCl , LaOI , SrHBr , SrI_2 , TlF , and YOBr . These materials feature leakage current densities ranging from 10^{-5} A/cm^2 to 10^{-19} A/cm^2 and an EOT ranging from 0.05 nm to 0.5 nm”.

- In addition to the changes made in response to Reviewer 1, we changed Fig. 4 and its captions, with the new bilayer estimates :

Fig. 4. Leakage current and EOT for monolayers (green circles), bilayers (red squares), and trilayers (grey triangles) in n-MOS applications. Reference dielectrics are also shown: The blue star shows monolayer h-BN. The dark purple line shows the EOT corresponding to the different thicknesses of HfO_2 (between 1-10 nm), while the orange line represents 0.4 nm of interfacial SiO_2 in addition to HfO_2 . The shaded light grey area shows values that fall outside of the IRDS leakage current (per pitch) criterion of $100 \text{ pA}/\mu\text{m}$ for a transistor with a 28 nm pitch, an 18 nm long gate and an effective gate width of 107 nm [77]. The acceptable current density becomes 0.145 A/cm^2 . For the bilayer, we estimate the performance using the dielectric constant of monolayer (filled squares) and using the bulk dielectric constant (hollow squares).

Reviewer: III) With respect to the strongly (in plane) anisotropic screening properties of black phosphorus, it might be interesting to also give ϵ_x and ϵ_y separately and not just their average.

Author reply: We thank the reviewer for raising this point. Only 3 materials (AlOCl , InOCl , and ScOBr) out of 32 materials in our list show anisotropic behavior ($\epsilon_x \neq \epsilon_y$). The dielectric constants of the aforementioned materials with anisotropic behavior are now added to the Table S8 of the supplementary as below:

We added “For category 1c, the dielectric constants in the plane are anisotropic and values are provided in the supplementary table S8.” in the main manuscript.

Table S8. Dielectric constant values for anisotropic materials.

		Material	Bulk (ϵ_∞)		Bulk (ϵ_0)		Monolayer (ϵ_∞)		Monolayer (ϵ_0)	
			x	y	x	y	x	y	x	y
Cat. 1	c	AlOCl	3.3	2.8	10.5	5.1	3.3	2.8	10.5	5.1
		InOCl	4.1	3.6	10.3	7.6	4.0	3.5	10.2	7.5
		ScOBr	4.7	3.7	14.4	9.4	4.7	3.7	14.3	9.3

Reviewer: IV) For the “vacuum elimination” the layer-layer distance is needed. Again, it is not clear how this has been evaluated, i.e. which exchange-correlation functional has been used and how reliable this is.

Author reply: We created a bilayer for every single material and the layer-layer distance is extracted from the inter-layer distance of the created bilayers as indicated in Fig. 1 [34]. For the exchange-correlation functional, we used vdW correction, DFT-D3 method of Grimme et al, as referenced in [82]. In addition. It is well established that the VdW correction is one of the most effective and reliable methods of DFT to study both short and long-range interactions.

REVIEWER COMMENTS

Reviewer #1 (Remarks to the Author):

The authors have addressed my concerns which I raised previously. However I have one more concern here. Sorry for I did not realize this problem in the first round.

Defects in materials are unavoidable during the synthesis process, and thus a single-layer material can hardly be the only dielectric, as one defect will lead to the degeneration. Thus, at least one more layer is needed, even under the best circumstance that the defect density is low enough, avoiding any two defects at the same xy position. Thus, in my opinion, single-layer need to meet the insulativity requirement, and double-layer need to meet the EOT requirement.

For these materials the insulativity is generally excessive, but the EOT for double-layer is a bit high for most materials, as the standard should not only be the leakage-qualified HfO₂+SiO₂ dielectric: ~0.65nm (calculation) in Fig. 4, but also a prospective number in IRDS of 0.50 nm (slightly better than the dielectric actual in use, in near-term challenges 2019-2024, IRDS Beyond CMOS, 2020). Thus, except for a modification of showing style in Fig.4, a new screening of is thus needed, and the conclusion of this article that which materials are promising, may be greatly changed. To take full advantage of the excessive insulativity, further applications for storage devices (leakage under DRAM limit, while EOT requirement is looser) may be a good choice for these materials.

Reviewer #3 (Remarks to the Author):

The authors answered to all of my comments, remarks, and questions. While some of these responses and the corresponding revisions are satisfactory, some of them have raised further questions and comments. Especially the unstable behaviour of ϵ_0 from Tab. S7 is alarming to me and should be explored and explained in more detail. However, by doing so and answering all the other minor questions from below, I would consider the manuscript to be ready for publication in Nature Communications.

In response to my first remark 1.) the authors made some efforts to increase the accessibility of their manuscript for non-experts, which should indeed help the latter.

In response to my second remark 2.A) on the treatment of long-range Coulomb interactions the authors write that the HSE06 functional is only used to calculate "the material's affinity and bandgap, not the dielectric constants or interlayer distance." and that VASP's DFPT implementation currently doesn't allow for HSE06 based calculations. Furthermore, they state that the long-wavelength ($q \rightarrow 0$) dielectric constants, listed here, do not suffer drastically from possibly missing long-range Coulomb interactions, as they show via vacuum-distance interpolated DFPT calculations.

To be precise: long-range Coulomb interactions *within* layered materials can affect both, the band gap (and the band structure in general) and the resulting screening properties. The first point might be captured via a fair comparison of HSE06 calculations to GW or similar results.

While I understand the technical difficulties arising from the available ab initio DFPT techniques, the authors missed my question on the accuracy of the HSE06 w.r.t the calculated and discussed band-gaps from Fig. 3. Without further comparisons the validity of HSE06 calculations is here unclear. I suggest to compare these results (or some of them) either with experimental results (which can also be flawed due to possible substrates), with available G0W0 calculations (such as listed in the Computational 2D Materials Database; <https://doi.org/10.1088/2053-1583/aacfc1> and accessible via <https://cmrdb.fysik.dtu.dk/c2db>), or with self-performed G0W0 calculations (using VASP this is not a major challenge anymore). Furthermore (and also w.r.t to my remark 2.D)) I would suggest to add the PBE results to Fig. 3 in order to have a homogeneous data basis to compare Fig. 3 with Tab. 1 and Fig. 2.

The author's comment on the impact of the long-range Coulomb interaction to the long-wavelength ($q \rightarrow 0$) dielectric constants and the new data from Tab. S7 are not entirely clear to me.

Conventional RPA calculations of the static macroscopic dielectric constant ϵ_∞ (as, for example, performed in VASP as part of GW calculations) for monolayer materials in three-dimensional cells are expected to significantly depend on the vacuum distance (especially in the vacuum ranges used in Tab. S7). Here, however, we do not see any effect at all. Now, I wonder, if this results from some applied extrapolation (such as Eq. (2) from the manuscript), the DFPT-based calculation of ϵ_∞ , or from the utilized PBE functional?

My remark B-1) has been perfectly answered by adding further references.

To reply to my comment B-2) on the monolayer stability, the authors added Gamma-point DFPT phonon calculations for all materials, which is greatly appreciated. From this one would expect TlF, GeClF, and actually also PbClF to be unstable in the free-standing monolayer limit. TlF is, however, still listed as a promising candidate without hinting towards its possible instability. PbClF, in turn, is not mentioned as an unstable monolayer material, although Tab. 9 lists imaginary frequencies for it.

In case the latter is not just a typo, my remark C) (first one) should be reconsidered.

The author's answer to my comment D) is not very convincing to me. The approximation " $\epsilon \sim 1/E_g$ " is certainly just an approximation and the exact charge density certainly also plays a major role. This, however, doesn't render the " $\epsilon \sim 1/E_g$ " behavior and the impact of the functional negligible. In this context, it would also be interesting to understand how the authors derive at the conclusion "while the band gap shows large differences between HSE and PBE+Grimme, the charge density shows a much smaller difference."? Finally, the comparison of the experimental and theoretical dielectric constants of CdBr₂, CdCl₂, and PbI₂ yielding deviations around 20% is certainly reassuring for these cases, but doesn't define the accuracy for all the other materials. However, I also understand that the available DFPT algorithms do not allow for such careful comparisons (although the impact of the functional to ϵ_∞ could be checked with VASP using other algorithms). Hence, for a coherent comparison between ϵ_∞ and ϵ_0 on the DFPT level the authors did all they can do at this point.

As a reply to the first referee as well as to my remark C) (the second) on the vacuum interpolation (I'm very sorry for not supplying the full second reference) of the ϵ results, the authors added Tab. S7 to the supplement, which shows that the out-of-plane component of ϵ_0 monolayer LaOCl is highly unstable w.r.t. to the vacuum distance. Correspondingly, the authors also added a statement on this to the method section stating that this "is not entirely surprising as the dielectric constant rescaling procedure amplifies errors, and phonon-related quantities, especially those involving polarization vectors, are known to have much larger errors compared to the ground state [86]". Although the authors also show that this is not an issue for LaOBr, this behaviour is highly alarming. Without further benchmarks and/or further detailed explanations it is not clear which monolayer ϵ_0 values from Tab. 1 and Fig. 2 are reliable and which are not. Thus, I would strongly suggest to check at least those materials which show extraordinarily high ϵ_0 as well as those which are in the list of the final most promising candidates. Furthermore, I would suggest to exclude all materials, which show an unstable behaviour like that, from the data presented and discussed in the main text.

My minor points I), III) and IV) have been satisfactorily taken care of / answered.

As a reply to the first referee and to my second minor point II) Fig. 4 has been strongly revised. Overall, this figure improved a lot and is now easier to access. I just have two minor follow up questions:

- a) How exactly is the EOT calculated? This should be explained in the methods.
- b) How exactly are the out-of-plane effective masses for the mono- and bilayer calculated? Or are the in-plane masses used? If so, this should be clearly stated and explained why this is a valid procedure.

Finally, after reading the manuscript again I realized that for a non-expert the difference between ϵ_∞ and ϵ_0 might not be that clear. It might be worth it to clearly define them, e.g. in the method section.

We thank the reviewers for their time and consideration. Below is the second point-by-point response addressing the concerns and the corresponding changes in the manuscript.

Reviewer #1

Reviewer: The authors have addressed my concerns which I raised previously. However I have one more concern here. Sorry for I did not realize this problem in the first round.

Defects in materials are unavoidable during the synthesis process, and thus a single-layer material can hardly be the only dielectric, as one defect will lead to the degeneration. Thus, at least one more layer is needed, even under the best circumstance that the defect density is low enough, avoiding any two defects at the same xy position. Thus, in my opinion, single-layer need to meet the insulativity requirement, and double-layer need to meet the EOT requirement.

For these materials the insulativity is generally excessive, but the EOT for double-layer is a bit high for most materials, as the standard should not only be the leakage-qualified HfO₂+SiO₂ dielectric: ~0.65nm (calculation) in Fig. 4, but also a prospective number in IRDS of 0.50 nm (slightly better than the dielectric actual in use, in near-term challenges 2019-2024, IRDS Beyond CMOS, 2020).

Thus, except for a modification of showing style in Fig.4, a new screening of is thus needed, and the conclusion of this article that which materials are promising, may be greatly changed. To take full advantage of the excessive insulativity, further applications for storage devices (leakage under DRAM limit, while EOT requirement is looser) may be a good choice for these materials.

Author reply: We thank the reviewer for their favorable comments and greatly appreciate raising this point.

We agree with the reviewer and are aware of the difficulty of avoiding defects in materials. It took a heroic effort to get silicon to its almost defect-free quality we know today. While underreported, defects have plagued new materials like TMDs and as the reviewer points out they may also be quite an issue in our newly proposed materials. How easily the proposed materials can be grown remains to be seen through further investigations but defects will likely appear in practice.

We think the reviewers idea that a “double-layer is needed to meet the EOT requirement” is interesting, it would limit the materials of interest only to LaOBr and LaOCl. But ultimately in practice, it could be too pessimistic, and we believe we should focus on what is possible instead of focusing on what may be impossible. The impact of defects on these dielectrics are not studied. Of course, a pin hole filled with metal would be dramatic and result in unacceptable leakage. But arguably, some defects could increase the barrier height and would thus not result in increased leakage. A detailed investigation of defects is beyond the scope of our paper but we think it is premature to rule out some of the other proposed materials.

In the conclusion we added:

“Monolayer dielectrics may not be sufficiently robust to defects and in this case, only LaOBr and LaOCl show sub-0.5 nm EOT in their bilayer forms”.

Reviewer #3

Reviewer: The authors answered to all of my comments, remarks, and questions. While some of these responses and the corresponding revisions are satisfactory, some of them have raised further questions and comments. Especially the unstable behaviour of ϵ_0 from Tab. S7 is alarming to me and should be explored and explained in more detail. However, by doing so and answering all the other minor questions from below, I would consider the manuscript to be ready for publication in Nature Communications.

Author reply: We thank the reviewer for their positive comment and careful review, which assisted to improve the manuscript.

In response to my first remark 1.) the authors made some efforts to increase the accessibility of their manuscript for non-experts, which should indeed help the latter.

Author reply: We thank the reviewer again for their comments, increasing the readability for future readers of our paper. We do note that we had to shorten the abstract to 150 words because of editorial concerns and had to remove some of the verbiage we previously inserted in the abstract.

In response to my second remark 2.A) on the treatment of long-range Coulomb interactions the authors write that the HSE06 functional is only used to calculate “the material’s affinity and bandgap, not the dielectric constants or interlayer distance.” and that VASP’s DFPT implementation currently doesn’t allow for HSE06 based calculations. Furthermore, they state that the long-wavelength ($q \rightarrow 0$) dielectric constants, listed here, do not suffer drastically from possibly missing long-range Coulomb interactions, as they show via vacuum-distance interpolated DFPT calculations.

To be precise: long-range Coulomb interactions *within* layered materials can affect both, the band gap (and the band structure in general) and the resulting screening properties. The first point might be captured via a fair comparison of HSE06 calculations to GW or similar results.

While I understand the technical difficulties arising from the available ab initio DFPT techniques, the authors missed my question on the accuracy of the HSE06 w.r.t the calculated and discussed band-gaps from Fig. 3. Without further comparisons the validity of HSE06 calculations is here unclear. I suggest to compare these results (or some of them) either with experimental results (which can also be flawed due to possible substrates), with available G₀W₀ calculations (such as listed in the Computational 2D Materials Database; <https://doi.org/10.1088/2053-1583/aacfc1> and accessible via <https://cmrdb.fysik.dtu.dk/c2db>), or with self-performed G₀W₀ calculations (using VASP this is not a major challenge anymore). Furthermore (and also w.r.t to my remark 2.D)) I would suggest to add the PBE results to Fig. 3 in order to have a homogeneous data basis to compare Fig. 3 with Tab. 1 and Fig. 2.

Author reply: We thank the reviewer for suggesting the C2MB database and were unaware that it contained G₀W₀ data. In Table S5 we added the G₀W₀ calculation from the reference above for comparison. We think this will be a valuable addition while not putting us through the process of repeating all our calculations using GW calculations. However, some of the materials under study are not listed in the database (shown by “-” in Table S5). Among the available materials, G₀W₀ data for some of the materials in this study is not reported (shown by “-” Table S5) and only the G₀W₀ data of Cat. 2 with the same space group of our study is reported which we listed in Table S5. We also found the experimental bandgaps of monolayer for two materials which are added to Table S5.

The following sentence is added to the Result and Discussion section:

“Even larger predicted G₀W₀ band gaps obtained from other theoretical studies [50] along with the experimental band gaps for some 2D monolayers [51, 52] are included in Table S5”.

We did not follow the suggestion to add the PBE results to the main paper. We agree that from a computational consistency point of view, it would make sense and understand the reviewer's thinking. However, for experimental readers, having these values in the main paper would create confusion and paint a misleading picture on what should be experimentally expected for these materials.

Table S5. Monolayer band gaps from different methods. Band gap values by PBE and HSE are calculated in this work and by G_0W_0 are taken from [S4]. The symbol "--" indicates that the material is not listed in [S4] and the symbol "-" shows that the material is available but the G_0W_0 band gap is not reported for that material. Only the G_0W_0 data of materials in Cat. 2 with the same space group of our study are available in [S4].

	Material	PBE (eV)	HSE (eV)	G_0W_0 (eV)	Experimental (eV)	
Cat. 1	a	TlF	3.67	4.75	--	--
	b	CaHBr	4.18	5.44	-	-
		CaHI	3.82	4.87	--	--
		GeClF	2.91	3.96	--	--
		HoOI	3.55	4.52	--	--
		LaOBr	4.05	5.68	--	--
		LaOCl	4.24	5.83	--	--
		LaOI	3.40	4.81	--	--
		LuOBr	4.48	5.79	--	--
		LuOI	3.27	4.24	--	--
		NdOI	3.74	4.72	--	--
		PbClF	3.52	4.60	--	--
		SrBrF	5.32	6.57	--	--
		SrHBr	4.29	5.42	-	-
		SrHI	3.97	5.03	--	--
		YOBr	4.68	6.00	--	--
	c	AlOCl	5.86	7.38	-	-
		BiOCl	2.75	3.74	-	3.60 [S5]
		InOCl	2.62	4.05	-	-
ScOBr		3.27	4.82	-	-	
Cat. 2	CaI ₂	3.83	4.61	6.92	-	
	CdBr ₂	3.25	4.47	5.48	-	
	CdCl ₂	3.91	5.32	6.72	-	
	MgBr ₂	4.79	6.01	7.72	-	
	MgCl ₂	6.01	7.12	9.60	-	
	MgI ₂	3.62	4.61	5.74	-	
	PbI ₂	2.58	3.32	3.22	2.47 [S5]	
	ZnBr ₂	3.45	4.77	5.86	-	
	ZnCl ₂	4.49	6.04	7.44	-	
Cat. 3	PbF ₄	2.49	4.24	-	-	
	SnF ₄	3.86	6.05	-	-	
Cat. 4	SrI ₂	3.99	5.01	--	-	

The author's comment on the impact of the long-range Coulomb interaction to the long-wavelength ($q \rightarrow 0$) dielectric constants and the new data from Tab. S7 are not entirely clear to me. Conventional RPA calculations of the static macroscopic dielectric constant ϵ_∞ (as, for example, performed in VASP as part of GW calculations) for monolayer materials in three-dimensional cells are expected to significantly depend on the vacuum distance (especially in the vacuum ranges used in Tab. S7). Here, however, we do not see any effect at all. Now, I wonder, if this results from some applied extrapolation (such as Eq. (2) from the manuscript), the DFPT-based calculation of ϵ_∞ , or from the utilized PBE functional?

Author reply: Indeed, the absence of any thickness dependence is the result of the extrapolation outcome from Eq. 2. This is also explained in Nano letters 20.2 (2019): 841-851. The dielectric response is measured by the response of the charge to an applied displacement field. If a field is applied in the perpendicular direction, a dipole forms in the perpendicular response and as we wrote in our previous response, that dipole do not have a long-range kernel. I think the situation is different in GW because $q \neq 0$ dipoles form, which do have a long-range kernel. An in-depth discussion of this is beyond the current paper though.

My remark B-1) has been perfectly answered by adding further references.

To reply to my comment B-2) on the monolayer stability, the authors added Gamma-point DFPT phonon calculations for all materials, which is greatly appreciated. From this one would expect TlF, GeClF, and actually also PbClF to be unstable in the free-standing monolayer limit. TlF is, however, still listed as a promising candidate without hinting towards its possible instability. PbClF, in turn, is not mentioned as an unstable monolayer material, although Tab. 9 lists imaginary frequencies for it.

In case the latter is not just a typo, my remark C) (first one) should be reconsidered.

Author reply: We apologize the review for the typo in the PbClF phonon energies listed in Table S9 and thank you for capturing this typo. Accordingly, we revised the PbClF values in the updated version of our manuscript.

In addition, GeClF, as you mentioned, shows unstable behavior. Similarly, based on phonon energies TlF seems to be unstable too. Therefore, we updated our abstract by deleting TlF from our final list of candidates and introducing six candidates instead of seven candidates. We accordingly updated the conclusion of our manuscript as follow by removing TlF from the final list.

“Since hydrobromides are generally hygroscopic materials and TlF monolayers were found to be unstable, we exclude SrHBr and TlF from the shortlist of candidate ended up with six promising vdW dielectrics: ...”

The author's answer to my comment D) is not very convincing to me. The approximation “ $\epsilon \sim 1/E_g$ ” is certainly just an approximation and the exact charge density certainly also plays a major role. This, however, doesn't render the “ $\epsilon \sim 1/E_g$ ” behavior and the impact of the functional negligible. In this context, it would also be interesting to understand how the authors derive at the conclusion “while the band gap shows large differences between HSE and PBE+Grimme, the charge density shows a much smaller difference.”? Finally, the comparison of the experimental and theoretical dielectric constants of CdBr₂, CdCl₂, and PbI₂ yielding deviations around 20% is certainly reassuring for these cases but doesn't define the accuracy for all the other materials.

Author reply: Our response was formulated based on general discussions we had with colleagues, the general idea that eigenvalues are more sensitive to modifications in a Hamiltonian compared to wavefunctions, and that materials like HfO₂ do have a large dielectric constant while having a small gap. In researching our response to this comment of the reviewer we came across Phys. Rev. B 49, 5323

“Density-functional theory of the dielectric constant: Gradient-corrected calculation for silicon” which discusses the (lack of) importance of the gap problem for the calculation of the dielectric constant. The values calculated for silicon are not underestimated using LDA/Gradient functionals by a similar factor as the band gap.

We agree that the few experimental dielectric values that we found do not “prove” the accuracy of all our calculations, ultimately only experiments can. It is encouraging that all materials that we could find agree to the extent they do. Overall, we follow a consistent calculation methodology for all materials and our results agree with all experimentally available data.

However, I also understand that the available DFPT algorithms do not allow for such careful comparisons (although the impact of the functional to ϵ_∞ could be checked with VASP using other algorithms). Hence, for a coherent comparison between ϵ_∞ and ϵ_0 on the DFPT level the authors did all they can do at this point.

Author reply: We thank the reviewer for understanding.

As a reply to the first referee as well as to my remark C) (the second) on the vacuum interpolation (I’m very sorry for not supplying the full second reference) of the ϵ results, the authors added Tab. S7 to the supplement, which shows that the out-of-plane component of ϵ_0 monolayer LaOCl is highly unstable w.r.t. to the vacuum distance. Correspondingly, the authors also added a statement on this to the method section stating that this “is not entirely surprising as the dielectric constant rescaling procedure amplifies errors, and phonon-related quantities, especially those involving polarization vectors, are known to have much larger errors compared to the ground state [86].”. Although the authors also show that this is not an issue for LaOBr, this behaviour is highly alarming. Without further benchmarks and/or further detailed explanations it is not clear which monolayer ϵ_0 values from Tab. 1 and Fig. 2 are reliable and which are not. Thus, I would strongly suggest to check at least those materials which show extraordinarily high ϵ_0 as well as those which are in the list of the final most promising candidates.

Furthermore, I would suggest to exclude all materials, which show an unstable behaviour like that, from the data presented and discussed in the main text.

Author reply: To quantify our arguments, we calculated $\frac{d\epsilon_{2D}}{d\epsilon_{sc}} = \frac{c}{t} \frac{\epsilon_{2D}^2}{\epsilon_{sc}^2}$ for the out-of-plane direction, measuring the impact of the error of the supercell (VASP) calculation on the extracted 2D calculation. We reported the $\frac{d\epsilon_{2D}}{d\epsilon_{sc}}$ for LaOBr and LaOCl in both out-of-plane and in-plane directions in Table S7 in the supplementary. For LaOCl, $\frac{d\epsilon_{2D}}{d\epsilon_{sc}}$ exceeds 10^4 while for LaOBr, $\frac{d\epsilon_{2D}}{d\epsilon_{sc}} = 4 \times 10^2$. This means that the VASP calculation of epsilon needs to be about two orders of magnitude better for LaOCl compared to LaOBr. Since $\frac{d\epsilon_{2D}}{d\epsilon_{sc}}$ scales with ϵ_{2D}^2 and inspecting Table 1, LaOBr is the third most sensitive material (the second being PbClF with very similar sensitivity as LaOBr). Since we found that LaOBr is reliable, all materials except LaOCl should be stable.

We added the following sentences to our Results and Discussion section:

“The sensitivity of the extracted dielectric constant to the one calculated from DFT can be quantified by calculating $\frac{d\epsilon_{2D}}{d\epsilon_{sc}}$. In the out-of-plane direction $\frac{d\epsilon_{2D}}{d\epsilon_{sc}} = \frac{c}{t} \frac{\epsilon_{2D}^2}{\epsilon_{sc}^2}$ while in the in-plane direction $\frac{d\epsilon_{2D}}{d\epsilon_{sc}} = \frac{c}{t}$. For LaOBr and LaOCl, we reported the $\frac{d\epsilon_{2D}}{d\epsilon_{sc}}$ in the out-of-plane and in-plane directions in Table S7 in the supplementary file. Since $\frac{d\epsilon_{2D}}{d\epsilon_{sc}}$ scales with ϵ_{2D}^2 , LaOCl is orders of magnitude more sensitive to error propagation.”

Table S7. The sensitivity of dielectric constants to the one calculated from DFT for the monolayer of LaOBr and LaOCl for different vacuum sizes.

	Vacuum (Å)	$\frac{d\varepsilon_{2d}}{d\varepsilon_{sc}}$	
		\perp	\parallel
LaOBr	15	2.91×10^2	3.25
	25	3.88×10^2	3.89
	30	4.99×10^2	4.53
	35	5.84×10^2	5.17
LaOCl	15	6.02×10^3	3.63
	25	1.34×10^4	4.69
	30	7.13×10^3	5.45
	35	2.84×10^4	6.21

My minor points I), III) and IV) have been satisfactorily taken care of / answered. As a reply to the first referee and to my second minor point II) Fig. 4 has been strongly revised. Overall, this figure improved a lot and is now easier to access. I just have two minor follow up questions:

- a) How exactly is the EOT calculated? This should be explained in the methods.

Author reply: Thank you for raising this point. To explain the EOT calculation procedure, we updated the method section as below and added the following sentence:

“We also calculate $EOT = \left(\frac{\varepsilon_{SiO_2}}{\varepsilon_{dielectric}} \right) t_{dielectric}$ to easily compare the performance of various dielectric materials.”

- b) How exactly are the out-of-plane effective masses for the mono- and bilayer calculated? Or are the in-plane masses used? If so, this should be clearly stated and explained why this is a valid procedure.

Author reply: The effective mass used in our calculation are the out of plane effective mass. The following sentences are revised to explain the effective mass calculation process:

“To estimate leakage current, we calculate the out-of-plane effective masses from the energy dispersion diagram (E - K) across the conduction band minimum (for the electron effective mass) or the valence band maximum (for the hole effective mass). We extract the effective mass by computing a 100 k-point path in the bulk band structure using the PBE functional. The k-point path traverses the Brillouin zone in the out-of-plane direction and is chosen to start at the band extremum and to end at the Brillouin zone edge.”

Finally, after reading the manuscript again I realized that for a non-expert the difference between ϵ_{∞} and ϵ_0 might not be that clear. It might be worth it to clearly define them, e.g. in the method section.

Author reply: We appreciate raising this point. We expanded our discussion in subsection V.B.1 of the method section by revising the following sentence:

“From VASP, we extract both optical and static dielectric constants. The optical dielectric constant (ϵ_{∞}) represents the high-frequency response where only the electrons can respond to an applied electric field. The static dielectric constant (ϵ_0), on the other hand, represents the low-frequency response where both the electrons and ions can respond [66].”

Format Compatibility

We have made some changes to make our manuscript compatible with the standard format of Nature Communications journal. We added five sections (“Data availability”, “Code availability”, “Author contributions”, “Competing interests”, and “Additional information”). we shorten our abstract. We removed some of our references. Below, have listed all the changes we made into our manuscript.

Following sections have been added:

“Data availability

The data that support the findings of this study are available from the corresponding author upon reasonable request.”

“Code availability

The codes that are necessary to reproduce the findings of this study are available from the corresponding author upon reasonable request. All DFT calculations were performed by using the Vienna ab-initio simulation package (VASP).”

“Author contributions

W.G.V. conceived the project. M.R.O. performed the simulations. M.R.O., M.L.V.de.P., A.S., and W.G.V. analyzed the obtained results. M.R.O., M.L.V.de.P., A.S., and W.G.V. wrote the paper with contributing to the discussion and preparation of the manuscript.”

“Competing interests

The authors declare no competing interests.”

“Additional information

Supplementary information: The online version contains supplementary material available at ...”

According to the Nature Communications formatting, the abstract must be 150 words or fewer. We modified our abstract as follows:

“Abstract

To realize effective van der Waals (vdW) transistors, vdW dielectrics are needed in addition to vdW channel materials. We study the dielectric properties of 32 exfoliable vdW materials using first principles methods. We calculate the static and optical dielectric constants and discover a strong out-of-plane response in GeClF (11.0), LaOBr (13.2), LaOCl (55.8) and PbClF (15.2) monolayers, while the in-plane dielectric response is strong in BiOCl, PbClF, and TlF, ranging from 64.7 to 98.4. To assess their potential as gate dielectrics, we calculate the bandgap and electron affinity, and estimate the leakage current through the candidate dielectrics. We discover six monolayer dielectrics that promise to outperform bulk HfO₂: HoOI, LaOBr, LaOCl, LaOI, SrI₂, and YOBr with low leakage current and low equivalent oxide thickness. Of these, LaOBr and LaOCl are the most promising and our findings motivate the growth and exfoliation of rare-earth oxyhalides for their use as vdW dielectrics.”

According to the Nature Communications formatting, the number of references should be limited to 70. The following references have been removed:

- [5] R. M. More, "Surface roughness scattering of electrons in semiconductors and semimetals," *Journal of Physics C: Solid State Physics*, vol. 8.22, p. 3810, 1975.
- [8] N. Briggs and et al, "A roadmap for electronic grade 2D materials," *2D Materials*, vol. 6.2, p. 022001, 2019.
- [11] W. M. Weber and T. Mikolajick, "Silicon and germanium nanowire electronics: physics of conventional and unconventional transistors," *Reports on Progress in Physics*, vol. 80.6, p. 066502, 2017.
- [13] D. Lynnall and et al, "Surface state dynamics dictating transport in InAs nanowires," *Nano letters*, vol. 18.2, pp. 1387-1395, 018.
- [14] C. C. Cheng and et al, "First demonstration of 40-nm channel length top-gate WS₂ pFET using channel-area-selective CVD-growth directly on SiO_x/Si substrate," *2019 Symposium on VLSI Technology*, IEEE, 2019.
- [16] H. Li and et al, "Toward the growth of high-mobility 2D transition metal dichalcogenide semiconductors," *Advanced Materials Interfaces*, vol. 6.24, p. 1900220, 2019.
- [18] H. P. Komsa and e. al, "Two-dimensional transition metal dichalcogenides under electron irradiation: defect production and doping," *Physical review letters*, vol. 109.3, p. 035503, 2012.
- [21] H. L. Tang and e. al, "Multilayer graphene-WSe₂ heterostructures for WSe₂ transistors," *ACS nano*, vol. 11.12, pp. 12817-12823, 2017.

- [22] X. Duan and et al, "Two-dimensional transition metal dichalcogenides as atomically thin semiconductors: opportunities and challenges.," *Chemical Society Reviews*, vol. 44.24, pp. 8859-8876, 2015.
- [24] T. Schram and et al, "WS₂ transistors on 300 mm wafers with BEOL compatibility," 2017 47th European Solid State Device Research Conference (ESSDERC). IEEE, 2017.
- [26] S. McDonnell and et al, "HfO₂ on MoS₂ by atomic layer deposition: adsorption mechanisms and thickness scalability," *ACS nano*, vol. 7.11, pp. 10354-10361, 2013.
- [30] H. G. Kim and H. B. R. Lee, "Atomic layer deposition on 2D materials.," *Chemistry of Materials*, vol. 29.9, pp. 3809-3826, 2017.
- [33] T. Roy and et al, "Field-effect transistors built from all two-dimensional material components," *ACS nano*, vol. 8.6, pp. 6259-6264, 2014.
- [41] Wang, Binghao, et al, "High-k gate dielectrics for emerging flexible and stretchable electronics," *Chemical reviews* 118.11, pp. 5690-5754, 2018.
- [47] Chen, Jianming, et al, "A high-density inorganic scintillator: lead fluoride chloride," *Journal of Physics D: Applied Physics* 37.6, p. 938, 2008.
- [54] J. Hölsä and . P. Porcher, "Crystal field effects in REOBr: Eu³⁺," *The Journal of Chemical Physics*, vol. 76.6, pp. 2790-2797, 1982.
- [55] Hofstadter, R., E. W. O'Dell, and C. T. Schmidt, "CaI₂ and CaI₂ (Eu) scintillation crystals," *Review of Scientific Instruments* 35.2, pp. 246-247, 1964.
- [58] A. Ferreira da Silva and et al, "Optical determination of the direct bandgap energy of lead iodide crystals.," *Applied physics letters*, vol. 69.13, pp. 1930-1932, 1996.
- [59] M. Shikr and S. Alfaify, "Tailoring the structural, morphological, optical and dielectric properties of lead iodide through Nd³⁺ doping.," *Scientific reports*, vol. 7.1, pp. 1-9, 2017.
- [62] R. McKinney and et al, "Rapid Prediction of Anisotropic Lattice Thermal Conductivity: Application to Layered Materials," *Chemistry of Materials*, vol. 31.6, pp. 2048-2057, 2019.
- [64] V. Pankratov and et al, "Luminescence and ultraviolet excitation spectroscopy of SrI₂ and SrI₂:Eu²⁺," *Radiation measurements*, vol. 56, pp. 13-17, 2013.
- [91] . R. K. Chanana, "BOEMDET Band Offsets and Effective Mass Determination Technique utilizing Fowler-Nordheim tunneling slope constants in MIS devices on silicon," *IOSR Journal of Applied Physics*, vol. 6, pp. 55-61, 2014.

REVIEWERS' COMMENTS

Reviewer #1 (Remarks to the Author):

I recommend that this paper be accepted for publication.

Reviewer #3 (Remarks to the Author):

The authors mostly adequately answered my questions and mostly resolved the alarming unstable vacuum extrapolation of the out-of-plane monolayer dielectric constants. Based on the latter it is now clear that the LaOCl data is not reliable, while all other extrapolations seem to be stable. The situation for PbClF is, however, still not finally settled. Its behavior is just estimated without explicitly investigating it. Next to that, the authors also made clear that TlF monolayer are not stable.

Based on this, I do not understand why TlF, LaOCl, and PbClF are still highlighted in the abstract. LaOCl's unstable (and thus unreliable) out-of-plane dielectric constant is still mentioned in the abstract and the material is still on the list of most promising dielectrics in the conclusions. The ab initio calculations for these three materials are not reliable, so that their prominent presentation in the abstract and conclusion is misleading.

However, this seems to be easy to fix after which the manuscript seems to me ready for publication.

We thank all the reviewers for their time and consideration and accepting our manuscript for publication. Below is the final answer to the reviewer 3 along with the corresponding changes in the manuscript.

Reviewer #3

The authors mostly adequately answered my questions and mostly resolved the alarming unstable vacuum extrapolation of the out-of-plane monolayer dielectric constants. Based on the latter it is now clear that the LaOCl data is not reliable, while all other extrapolations seem to be stable. The situation for PbClF is, however, still not finally settled. Its behavior is just estimated without explicitly investigating it. Next to that, the authors also made clear that TlF monolayer are not stable. Based on this, I do not understand why TlF, LaOCl, and PbClF are still highlighted in the abstract. LaOCl's unstable (and thus unreliable) out-of-plane dielectric constant is still mentioned in the abstract and the material is still on the list of most promising dielectrics in the conclusions. The ab initio calculations for these three materials are not reliable, so that their prominent presentation in the abstract and conclusion is misleading.

However, this seems to be easy to fix after which the manuscript seems to me ready for publication.

Author reply: As the reviewer points out, the exact value of LaOCl is not reliable but it is clear that LaOCl does have the largest value so it should certainly be highlighted as a material of great interest in the abstract and the conclusion. For PbClF, the situation for PbClF is relatively clear since we quantified that the error in dielectric constants is proportional to ϵ^2 and ϵ^2 only differs by a factor of 1.32 between LaOBr and PbClF whereas the difference between LaOCl and LaOBr is orders of magnitude. This indicates that PbClF will have the error on the same order of magnitude as LaOBr and therefore be accurate. Concerning the mentioning of TlF, it is not stable in monolayer form but it is in bulk form. The reviewers' criticism is valid for the LaOCl dielectric constant value mentioned in the abstract, which is not accurate, and we have removed the values from the abstract.

Other changes

The following section were also updated:

Data availability

The input files used in this study have been deposited in the NOMAD repository database under accession code (<https://dx.doi.org/10.17172/NOMAD/2021.07.18-1>). The processed dielectric data are available in the main paper. The lattice constant, bandgap and calculated leakage current are provided in the Supplementary Information file.

Table 1 was updated to comply with formatting standards and the caption was updated:

Materials category 1a-4, illustrated in Fig. 1 and space groups provided in Supplementary Table 1, are indicated in parenthesis in the first column for each material.

Figure 1 caption was updated:

The structure of 32 vdW Materials. Side and top views of the monolayer structures are shown, where the yellow squares represent the computational unit cells. Side view of the bilayer structures are also demonstrated,